# Low Uncertainty Wave Tank Testing and Validation of numerical methods for Floating Offshore Wind Turbines

Christian W. Schulz[1], Stefan Netzband[1], Philip D. Knipper[1], and Moustafa Abdel-Maksoud[1]

[1]Hamburg University of Technology, Am Schwarzenberg-Campus 4, 21073 Hamburg, Germany

**Correspondence:** Christian W. Schulz (christian.schulz@tuhh.de)

**Abstract.** The accurate simulation of loads and motions of Floating Offshore Wind Turbines (FOWT) in operation is key to the commercialisation of this technology. To improve such load predictions, a critical assessment of the capabilities and limitations of simulation methods for FOWT is mandatory. However, uncertainties arise during the whole validation process of a numerical method. These can drastically impair the quality of the validation. In the case of FOWT, the interaction between aerodynamic, hydrodynamic and mooring loads on the one hand and platform motions on the other hand causes a high level of uncertainty in the measurement data acquired in model tests. This also applies to comparing a numerical model to the test data, as these interactions make the distinction between cause and effect challenging. To address these challenges, several improvements to the validation process aiming at the reduction of the uncertainties are proposed and evaluated in this work. The major improvements are the measurement of the rotor thrust force excluding the tower top inertia loads, a significant improvement of the wind field quality in the wave tank, a comparison of the rotor aerodynamics in wind tunnel and wave tank and the utilisation of hybrid simulations based on the measured platform motions. These steps are applied to wave tank tests of a FOWT utilising a single point mooring and the subsequent validation of the numerical panel method *pan*MARE. The improvements allowed for a considerable decrease in the random and systematic uncertainty of the model tests and made a valuable contribution to the distinction between cause and effect regarding the deviations between measurements and simulations.

## 1 Introduction

The ability to perform precise and reliable simulations of the motion behaviour of Floating Offshore Wind Turbines (FOWT) is a key requirement for developing resource- and cost-effective designs. Numerous methods for simulating FOWT dynamics with very different degrees of complexity have been developed. However, a critical assessment of the modelling inaccuracies and expected deviations from simulated to real loads of the various simulation approaches is necessary. A starting point for this validation process is the generation of measurement data in model tests. Here, the interaction between the hydrodynamic loads on the floating platform, the tension of the mooring lines and the aerodynamic loads acting on the rotor is reproduced in the wave tank. The obtained motions and loads can then be compared to simulation results to identify, quantify, and understand the observed deviations between the experiment and the simulation. Due to the high complexity of the FOWT motion dynamics, multiple challenges arise throughout the validation process, which could potentially impair the quality of the assessment of simulation methods.

**Motivation**

A major challenge is the generation of reliable and accurate measurement data. Considerable effort has been undertaken by the research community in the last two decades to provide suitable data sets for validation. However, a relatively high level of systematic and random uncertainty was observed in several experimental studies. This is especially an issue when the complete wind turbine is considered, and a wind field is applied in the wave basin. In this case, referred to as 'full physical testing', the generation of the wind field is a driver of the observed uncertainties.

In addition to the difficulties in practical testing, it turned out to be challenging to clearly link deviations between simulations and measurements to a specific modelling insufficiency of the utilised simulation method. Especially in the case of deviations between measurements and simulations that can be caused by multiple reasons, it is challenging to distinguish between the impacts of the different physical systems on certain loads or motions. As a consequence, it is difficult to clearly understand the physical reasons for such deviations and improve the simulation method accordingly. The measurement of interface loads at the tower top and mooring points can help to mitigate this problem. While this proved to be successful for the mooring loads, it remains challenging to validate aerodynamic loads using tower top measurements because inertial loads due to platform motion and tower vibrations superimpose the measurement signal.

The value of the validation results is also impacted by the choice of load cases as well as the comparison metrics between simulations and measurements. As many wave tank experiments do not exclusively serve as validation cases but also as practical tests for FOWT designs, irregular sea states are often utilised to investigate the FOWT under realistic conditions. In consequence, a time-domain comparison between measurements and simulations is often difficult to realise due to the complexity of these load cases. Frequency domain comparisons like amplitude and power spectral density spectra are suitable for comparisons regarding the general motion behaviour of the FOWT in these cases. However, without the consideration of the corresponding phase spectra, the transformation of experimental and numerical results in the frequency domain results in loosing a part of the information on the transient behaviour and interrelations between motions and loads. While statistical analyses are suitable for the characterisation of the overall motion behaviour, the quantification and identification of simulation inaccuracies arising from insufficient modelling can be challenging when such comparison metrics are used only.

**Scope**

This work aims at an improvement of the described validation process for FOWT simulation methods in order to gain better insight into the modelling deficits and the level of uncertainty that can be expected in FOWT simulations. Thus, a number of potential improvements for the quality of this process ranging from the experimental setup to the comparison metrics are proposed and evaluated. These are in particular:

*Reduction of systematic and random measurement uncertainty*

- The wind field quality is significantly increased in terms of flow non-uniformity and turbulence intensity. This is achieved by the use of an elaborate wind generator design. The new design leads to a larger size of the wind generator. Due to the

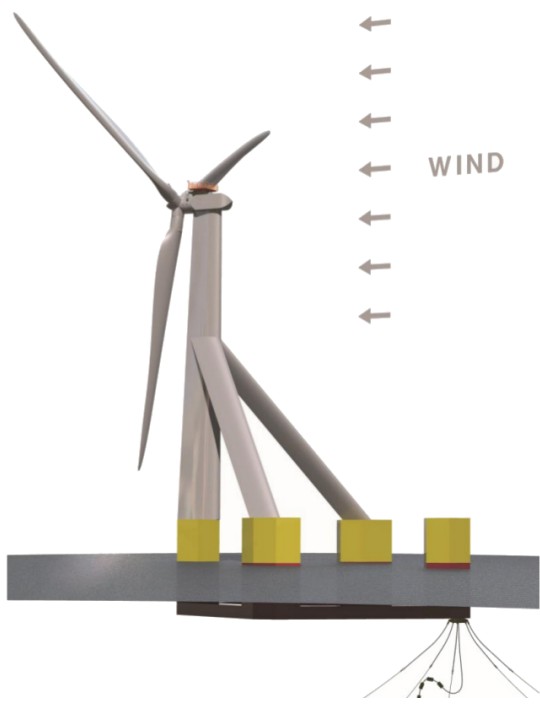

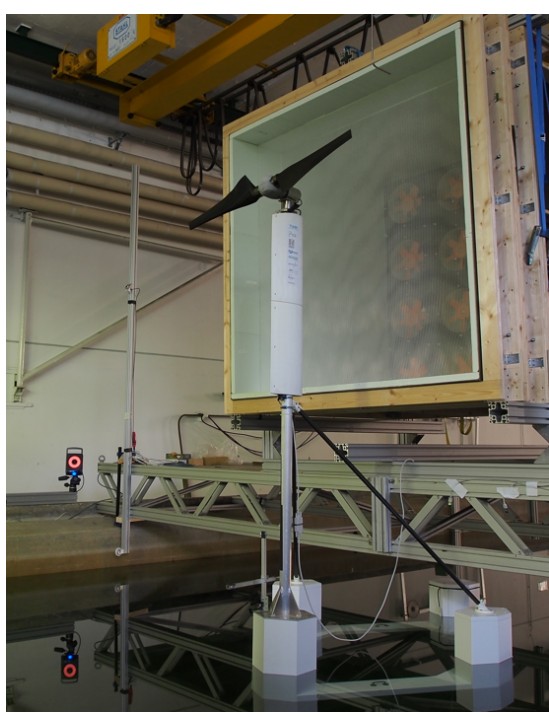

**Figure 1.** Illustration of the Curse Offshore SelfAligner. Reprinted with permission of CRUSE Offshore GmbH.

**Figure 2.** Photograph of the model setup.

limited space in the wave tank, the size of the wind field and, consequently, the wind turbine rotor needs to be reduced
while the wind speed is increased in order to achieve a Froude-scaled thrust force.

– A wireless data acquisition system is utilised to minimise the influence of the cable bundle on the platform motion. The land connection is reduced to a single cable with nearly negligible weight.

– Every measurement case is repeated at least twice in order to continuously monitor the repetition error.

*Enabling isolated validation of simulation sub modules*

– The reduced size of the rotor enabled a detailed investigation of the rotor thrust force in a wind tunnel environment to which the wave tank measurements are compared.

– Tower top loads are measured in all three directions, and a procedure to remove the inertial and gravitational loads is applied and validated to determine the aerodynamic thrust force in time domain. Due to the improved quality and a detailed characterisation of the wind field in combination with the measurement of the aerodynamic thrust, a clear
identification of the contribution of the aerodynamic loads to the platform motion is enabled.

– The mooring forces in all three directions are measured at the connection of the mooring to the floater.

– Prior to the full validation, a validation of the simulation sub-modules for mooring and aerodynamics is performed. This is realised by the application of the measured platform motion trajectory in the simulation model, which is referred to as 'hybrid simulation'. The results are directly compared to the measurements of the aerodynamic thrust force and the mooring loads. In this way, the rotor and mooring loads are validated under test conditions but isolated from possible insufficiency in the prediction of the platform motion. This concept has been applied successfully e.g. by Hall and Goupee (2015) in model-scale and by Netzband et al. (2023) in full-scale.

*Validation with phase-averaged time-domain measurements*

– Simple, periodic load cases are chosen to deliver basic validation data with high accuracy.

– Time-domain measurements are obtained using phase-averaging. In this way, a considerably low influence of random uncertainty on the comparison between measurements and simulations can be reached.

In this work, the above-proposed improvements are applied to the validation procedure of the first order panel method *pan*MARE and its lifting line sub-module using the Curse Offshore SelfAligner FOWT (see Figure 1), which is passive yaw concept. The wind turbine was downsized by a factor of 3 compared to the scaling of the platform, which yields a smaller wind generator cross-section. A major drawback of this procedure is the violation of the Froude similarity in the time scale of the wind field and the tower top motion. This violation causes a reduced sensitivity of the rotor loads on the tower top motions. Nevertheless, the dynamic effects of platform motions on the rotor loads and vice verca are measured and can be accurately modelled in the numerical method.

In the following sections, relevant parts of a number of previous studies are summarised to present the context of this work. Subsequently, the downsizing of the rotor is discussed, and the utilised numerical method, as well as the design of the wind generator, are introduced. Furthermore, the complete wave basin setup is depicted, a simplified uncertainty analysis is performed and a series of repetition tests is presented to quantify the expectable uncertainty. Measurement data is utilised to validate the modules of *pan*MARE and their coupling in hybrid and full simulations. Finally, the findings from the validation and the application of the proposed experimental technique are concluded.

## 2 Previous works in the field of wave tank testing and validation of numerical methods

A considerable number of experimental investigations on loads and motions of FOWT in model-scale have been performed in the past two decades. Otter et al. (2021), Murphy et al. (2015), Chen et al. (2020) and Gueydon et al. (2020) provide general reviews and an overview of most of these investigations in wave tanks. In these, different ways of considering the aerodynamic loading on the rotor in the context of the scaling issue between Froude similarity on the hydrodynamic side and Reynolds similarity on the aerodynamic side were evaluated. Apart from some early studies utilising a simple drag disk (e.g. Roddier et al. (2010)), the generation of aerodynamic loads in recent works can be divided into three categories: Bottom-fixed thrust generation, hybrid testing and full physical testing. The advantages and disadvantages of the first two approaches are briefly

described in the following, while a more detailed view is given on the full physical testing approach as it is most relevant for the present study.

The bottom fixed thrust generation offers a simple way to model at least the mean thrust force and fluctuations of the incoming wind. Here, a predefined thrust force time series is applied to the tower top with the aid of a controlled propeller (see e.g. Andersen (2016) and Desmond et al. (2019)). The major drawback of this technique is that there is no interaction between the aerodynamic loads and the tower top motion. Consequently, important coupling effects like aerodynamic damping cannot be reproduced in the wave tank tests. The bottom-fixed thrust force generation approach can therefore be considered as a cost-efficient way for the testing of new FOWT designs, but its applicability for the generation of validation data including aerodynamic loading is limited.

A considerable number of hybrid testing devices have been developed and tested in recent years. Otter et al. (2021) provide a detailed review of these devices. To date, two different types of actuators are utilised to apply the rotor loads to the platform. Either propellers or multiple cables connected to winches are mounted on the tower top. Both techniques utilise a real-time numerical simulation of the aerodynamic domain, which is coupled with the measured tower top motions of the platform model. Finally, a feedback loop is created, so that the tower top motions are considered in the aerodynamic simulation while the actual aerodynamic loads are applied to the wave tank model by the actuators. The suitability of the hybrid testing approach for wave tank testing of new FOWT prototypes and validation tests focusing on hydrodynamics was demonstrated in a number of studies (e.g. Azcona et al. (2014), Amaral et al. (2021), Otter et al. (2020), Hall and Goupee (2018), Vittori et al. (2022)). Naturally, a validation of the aerodynamic loads is not possible with this approach. In addition, there is still a lack of investigations on the influence of the time lag between desired and realised tower top loads on the motion behaviour, see Gueydon et al. (2020). Similarly, more investigations of the random uncertainty arising from the actuators are needed to characterise the capabilities and limitations of this approach.

**Full physical testing**

To date, full physical testing is an important option to gain validation data for numerical models covering the aerodynamic and hydrodynamic domain. In this approach, a wind generator is placed in the wave tank, and a model rotor is installed on the floating platform. The full physical testing approach is widely used and a number of examples can be found in the above mentioned reviews. In the following, relevant test campaigns and validation studies are briefly described in order to summarise the advantages and issues of this testing methodology.

In most full physical tests, the rotor is not geometrically scaled as special low Reynolds number airfoil shapes are used. In addition, the chord length of the blade sections is increased to achieve a minimum of similarity to the full-scale characteristic of a wind turbine rotor in terms of Reynolds number and blade loading. One of the early studies, the DeepCWind campaign led by the University of Maine in 2011, showed that the utilisation of geometric scaling of the blades lead to a significantly reduced thrust force and an extremely low power output, see Goupee et al. (2014). A redesign of the rotor with airfoils designed for low Reynolds numbers - as used in classical wind tunnel experiments - showed a considerable improvement of the rotor per-

formance and was used in a later test campaign. During the OC5 project (Offshore Code Comparison Collaboration Continued with Correlation) presented by Robertson et al. (2017), measurement data from this campaign was utilised to validate multiple numerical methods. As an uncertainty assessment was not performed during the tests, some consistent deviations between measurements and simulations were not fully understood. A high level of uncertainty in the generated wind field and/or its measurement, as well as the influence of the instrumentation cable bundle, were proposed as possible explanations for deviations between measurements and simulations. The quality of the wind field was indeed limited in these cases: Maximum spatial deviations from the mean wind speed up to 20 % at the lower border of the rotor swept area where measured, while deviations up to 10 % and turbulence intensities up to 7 % were present across the rest of the rotor swept area (Wendt et al., 2019). Wendt et al. (2019) calibrated a simulation model in the FAST simulation framework to match the results of a test campaign on the DeepCWind semi-submersible with the redesigned rotor. The calibration improved the agreement between simulations and measurements. However, issues in the statistically evaluated tower base force remained. A summary of the DeepCWind tests recommended a higher quality of the wind field, as strong non-uniformity was observed, and wireless data acquisition in order to reduce the influence of the cable bundle, see Robertson et al. (2013).

Bredmose et al. (2017) performed a full physical test of a novel platform design aiming at the investigation of aerodynamic damping and the influence of the wind turbine controller on the platform motion. A strong effect of aerodynamic damping on the platform pitch and surge response to a focused wave group was shown. In wind-only tests, the effect of negative damping (see Jonkman (2008)) due to the use of an onshore blade pitch controller was demonstrated, while the developed offshore controller did not excite the platform motion. In regular and irregular wave tests, the superior motion damping of the offshore controller could be proved. In a regular wave only condition, exemplary time domain comparisons with a simulation using FAST showed good agreement with the measurements. Flow measurements in the wind field revealed strong non-uniformities of up to 21 % of the mean wind speed, while Madsen et al. (2020) measured turbulence intensities ranging from below 10 % up to 20 % in the wind field of the same wind generator. In addition, a part of the lower rotor half was not entirely covered by the measurements. As a consequence, high-frequent noise appeared in the thrust force during constant wind tests. In the case of rotor torque, noise was even observed in low-frequency ranges. Finally, the application of the wind field yielded an increase of the scatter in the pitch motion compared to cases without wind.

Yu et al. (2017) presented a test series focusing on the implementation of a controller, including measurements of the tower top loads, using the same model. Steady measurements showed that the results of the aerodynamic simulation model needed to be adjusted by 65 % to match the measured power coefficient. In addition, a large oscillation of the rotor thrust force was monitored in the experiments, which resulted in the occurrence of negative values for the thrust force. Although named as aerodynamic thrust force, it is likely that the gravitational and inertia loads have not been compensated from the signal. Therefore, the tower top force rather than the rotor thrust is probably shown.

Cao et al. (2022) and Wang et al. (2024) also report the usage of an inertia removal procedure applied to tower top force measurements. However, no validation or verification of the procedures and no distinct comparison with aerodynamic simula-

tions is given in these works.

Very recently, 17 participants of the OC6 project (Offshore Code Comparison, Collaboration, Continued, with Correlation and unCertainty)[1], used model tests of the Tetraspar FOWT as a basis for a broad validation study with 15 different simulation methods (see Bergua et al. (2023)). The validation campaign showed a promising overall agreement between the measurements and the different simulation models. However, it was found that the cable bundle had a significant influence on the motion of the platform, and the simulations had to be corrected for this influence. Although tower top loads have been recorded, an explicit validation of aerodynamic loads under wave excitation has not been performed.

In summary, the full physical testing approach was applied successfully to various different FOWT concepts and provided insight into their motion dynamics. As an example, the presence of aerodynamic damping could be proved in several studies. However, general issues regarding the quality of the measurements and the wind fields were reported. For example, Robertson (2017) claimed that more repeat tests in wave tank testing of FOWTs are necessary to investigate the random uncertainty as a consequence of the DeepCWind measurement campaign. Unfortunately, publications containing consequent repetitions of the tests are still rare. Similarly, Gueydon et al. (2020) recommended accumulating more evidence on the validity of the applied testing methodology in the context of measurement uncertainties. Many works report issues on the quality of the wind field, although considerable effort was undertaken to manufacture elaborate wind generators in a large scale. A particular problem is the full coverage of the rotor swept area with a high-quality wind field as keeping the mandatory distance between the wavecrest and the wind generator often results in strong boundary effects in the wind field near the lower edge of the rotor swept area. In addition, investigations of the tower top loads have been performed in few studies, so the consequences of the quality issues in the wind fields are difficult to quantify. It is likely that a considerable portion of the random uncertainty observed in the above studies arises from undesired flow non-uniformity and high turbulence intensity in the wind field. A compensation of inertia loads to directly evaluate the rotor thrust or torque seems to be rare, if even present.

In most cases, comparisons of numerical and experimental results are performed on the basis of statistically evaluated amplitude spectra, which may not tap the full potential of the information gained from the experiments. As statistical values are comparably insensitive to noise, a possible reason for the choice of these evaluation methods could be that the random uncertainty in the measurements makes accurate time domain investigations challenging. In addition, the performed validation studies focus on simulations utilising frequency domain methods in terms of hydrodynamics. Comparing those results to measurements in frequency domain may yield better performance than can be expected in time domain because these methods use similar frequency domain results from higher fidelity methods as input.

---

[1]https://iea-wind.org/task30/, last accessed on 2nd April 2024

## 3 Introduction and discussion of the scaling approach

Previous studies showed that the design and manufacturing of large-size wind generators with high flow quality is an extremely
challenging task. Therefore, a reduction of the rotor size - resulting in a reduction of the wind generator size - is applied in
order to allow for a more elaborate wind generator design. In the following section, the consequences of this approach on the
similarity to the full-scale FOWT in terms of Froude similarity are discussed.

To maintain the similarity between the motion behaviour of the scale model and the full-scale FOWT, a similarity of gravi-
tational and inertial loads (Froude similarity) is primarily required. This is due to the fact that ocean waves and the rigid body
motion of a floating body are mainly driven by gravitational and inertia effects. However, usually, it is not possible to concur-
rently maintain Reynolds similarity due to the absence of a model fluid with a suitable kinematic viscosity, see Hughes (1993).
For most floating bodies, viscous forces like drag do not have a driving influence on the floating motion and can, therefore, be
disregarded, approving a certain error or artificially modelled in numerical models to match experimental data. In contrast to
this, viscous effects are most relevant for wind turbine aerodynamics. The violation of the Reynolds similarity leads to strong
deviations in the aerodynamic loads when geometrical scaling is used. This is due to the difference in the Reynolds numbers at
the blade sections compared to a full-scale turbine, which is extreme in this case. Therefore, downscaled rotor blades usually
need to be redesigned for the low Reynolds number regime in model-scale as already mentioned in section 2. In this way, a
thrust characteristic similar to that of a real wind turbine can be achieved, which fulfils the Froude similarity of the rotor thrust
force. However, in practice, the thrust characteristics of modern wind turbines are quite similar, so the redesigned turbine often
serves as a general representative for a certain power rating rather than reflecting an individual turbine. From this, it becomes
clear that the rotor is not necessarily a model of the full-scale version; however, it can be considered as an external substitute
system that provides suitable aerodynamic loads in terms of Froude similarity.

In the present case, a representative rotor designed for model experiments was utilisied instead of a truly downscaled version
of the full scale wind turbine for the reasons discussed above. The diameter of the model rotor is not scaled with the same factor
that is used for the platform as it would be required by Froude scaling laws. A photograph of the downscaled model is shown
in Figure 2. While the the platform and the hydrodynamic environment are scaled using conventional Froude scaling with a
scaling factor of $\lambda_{hydro} = 45$, the geometric scaling factor for the wind turbine rotor diameter was chosen to be $\lambda_{aero} = 150$,
as illustrated in Figure 3. In this case, the model rotor is considered as a subsystem, which generates similar thrust and torque
as an ideally scaled rotor would deliver. In order to achieve the Froude similarity of the mean thrust (assuming a constant thrust
coefficient and tip speed ratio), the wind speed needs to be increased by the factor $\lambda_{aero}/\lambda_{hydro}$ in comparison to conventional
Froude scaling of the environmental conditions. With this configuration, the reduced model rotor delivers the same mean rotor
thrust as an ideally scaled rotor in terms of Froude similarity (see appendix A.

As a consequence of the different scaling ratios of the rotor and platform, the kinematic similarity between the tower top
motion velocity due to pitch motion and the wind speed is violated. This leads to a reduction of the sensitivity of the rotor
thrust to the platform motion. For a harmonic surge motion, the thrust force amplitude is reduced by the factor $\lambda_{hydro}/\lambda_{aero}$
in comparison to the full-scale scenario. A detailed derivation is given in appendix A. In the present case, the amplitude is

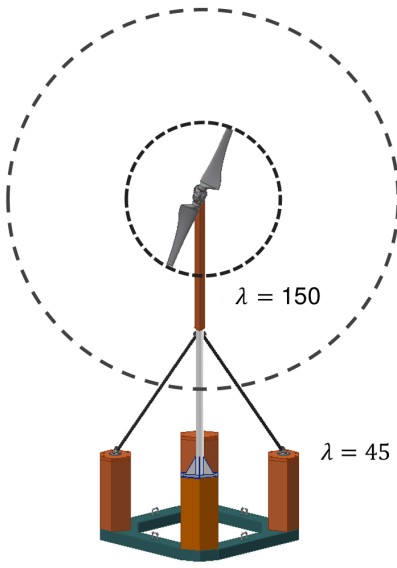

$\lambda = 150$

$\lambda = 45$

**Figure 3.** Sketch of the applied scaling ratios.

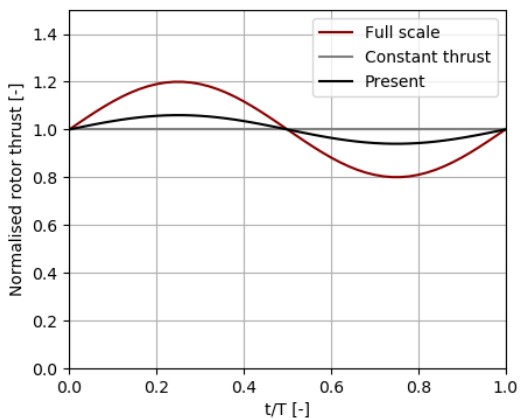

**Figure 4.** Illustration of the expected deviations in the normalised aerodynamic rotor thrust from full-scale to the present approach during an exemplary surge oscillation.

approximately reduced by a factor of 3. A comparison of the rotor thrust of the full-scale FOWT and the proposed scale model undergoing a sinusoidal surge motion is shown in Figure 4. As a result, the effect of aerodynamic and hydrodynamic interaction phenomena like aerodynamic damping is reduced; however, it is well-defined and can be observed in the platform motion and the tower top load measurements.

Although the Froude similarity of tower top motion velocity and wind speed is violated, the utility system is well suited for the validation of numerical methods. The special value of the proposed scaling approach is to enable a precisely known wind environment and wind turbine characteristic. This is achieved by a more elaborate wind generator and a much better covering of the (moving) rotor swept area compared to other tests due to the small size of the turbine. The ability to test the smaller rotor in a wind tunnel environment with sufficiently low blockage ratio and under highly controlled conditions (see section 7.1) opens the possibility to validate the aerodynamic simulation model accurately and to identify measurement differences arising from the non-ideal wave tank environment. Both together leads to a well defined (and known) thrust force, which is applied to the tower top with a comparatively low level of noise. This, in turn, yields a low contribution of the aerodynamic system to the random and systematic uncertainty of the platform motion, which is a major improvement in comparison to the above listed studies. In addition, the low level of noise enables the reliable compensation of gravitational and inertial loads from the tower top forces so that a direct validation of the simulated rotor thrust is possible during a motion cycle. Naturally, these improvements over existing test strategies are achieved in exchange with a less realistic behaviour of the model FOWT as discussed above.

## 4   Test setup and measurement system

The model tests were performed in the 5 m wide and 80 m long towing tank of the Hamburg University of Technology. A platform was mounted over the water to accommodate the wind generator. The scale model of the FOWT and the wind generator were placed in the first third of the wave tank and aligned with its centre line.

In this section, the wind generation system, the physical model, the used sensors and data acquisition system is described. Particular emphasis is put on the investigation of the wind generator performance and the acquisition of the inertia force 260 compensated rotor thrust force.

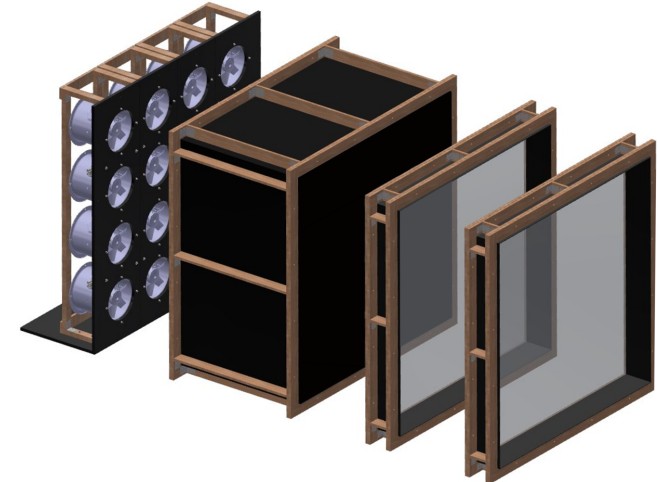

**Figure 5.** Illustration of the wind generator assembly.

### 4.1   Wind generator

A modular, lightweight and easy-to-install wind generator has been designed and manufactured (see sketch in Figure 5). With an outlet cross section of 1.6 m by 1.6 m, the rotor is more than fully covered. The wind generator consists of an array of 16 fans, a settling zone with a length of 1 m and two mesh screens at a distance of 0.2 m and 0.4 m to the outlet. The mesh 265 screens consist of a rectangular grid of polyamide threads with a diameter of 0.39 mm and have an overall porosity of 62 %. The modules (four fan rows, one settling zone, two screens) are mounted using a wooden frame and can be combined in any order and with additional elements. However, no modular division of the wind generator cross-section was undertaken in order to avoid the introduction of flow non-uniformity between such modules. All modules can be easily handled by two persons so that the wind generator can be assembled on the platform over the wave basin without the need for extra equipment.

The 16 fans are powered by a frequency inverter, which can be controlled in closed-loop control with a velocity measurement using a Prandtl probe. However, it turned out that the slip of the asynchronous motors stays constant after a short initial warm-up so that the control of the supply frequency via the inverter is sufficient to maintain a constant wind speed. The Prandtl probe was therefore utilised to monitor the actual wind speed. Air pressure, temperature, and humidity were recorded prior to every

test to monitor the air density and its influence on the wind turbine loads as well as on velocity measurements.

The generated wind field was investigated on a plane with a distance of 1.2 m to the outlet using a Prandtl probe, which was mounted on a frame that allowed for an exact positioning of the 102 measurement points. Every point measurement was performed for 10 s, and the mean wind speed, as well as the turbulence intensity, have been calculated. The signal was sampled with a frequency of 1.2 kHz and low pass filtered by 350 Hz in order to exclude measurement noise but still consider fluctuations induced by the fan blades. In Figure 6, average wind speed and turbulence intensity are illustrated from an interpolation
between the measurement points. For the average wind speed, a maximum deviation of approximately 1.5 % between a single measurement point and the mean wind speed inside the denoted rotor swept area was observed. A measure for the overall non-uniformity is given by the coefficient of variation of the spatial wind speed variation, which is approximately 1 %. The turbulence intensity in this region was found to be below 5 %. Even though the flow quality slightly decreases towards the boundaries of the wind field, a high homogeneity can be maintained even if the rotor undergoes small motions.

A comparison of the flow quality and construction with other wind generators is given in Table 1, where the flow quality measures are estimated by the procedures described in appendix B. From the table, significant advantages of the present wind generator especially regarding the flow non-uniformity can be concluded. Both, maximum spatial deviation and spatial coefficient of variation of the mean wind speed are more than five times lower than those of the other wind generators. However,
the given values need to be considered with care as the estimation of the non-uniformity measures from the available data in literature contains a number of shortcomings (e.g. reading from color maps or very different averaging periods) that might have a relevant impact on the estimated values. Another shortcoming of the evaluation of the wind speed quality is that flow velocities perpendicular to the main flow direction have not been measured (or evaluated) neither in literature nor in the present case.

Additional flow measurements of the present wind generator showed that the sensitivity of the average wind speed and turbulence intensity to a variation of the distance to the wind generator outlet (0.8 m up to 1.6 m) is very limited. Slight changes in the wind speed, i.e. fan frequency, also did not cause significant deviations in the wind field quality. When removing one mesh screen, both maximum flow non-uniformity and turbulence intensity increased significantly. Initially, baffle plates were
300 installed inside the settling zone in order to reduce potential rotation in the flow field. As a result, it was found that the maximum deviation from the mean wind speed increased drastically. This can be explained by inaccuracies in the baffle plate's geometry and positioning. Therefore, the baffle plates have been removed.

Apart from the size, the main differences in the wind generator design and configuration in comparison to similar devices
can be summarised as follows:

- The wind generator outlet cross section is nearly twice as wide and high as the rotor diameter.

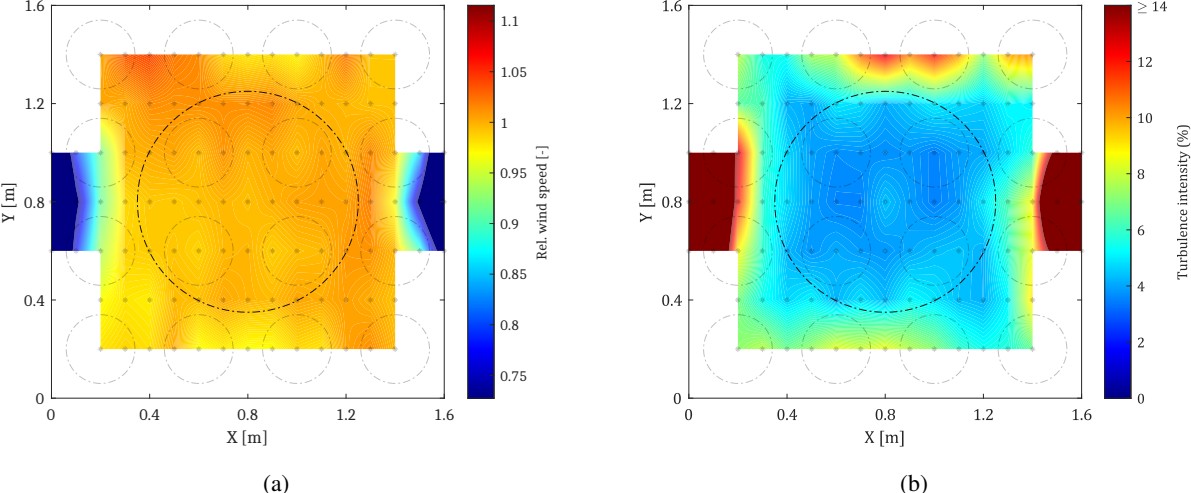

**Figure 6.** Velocity field measurements interpolated from from 102 measurement positions. (a) Normalised average wind velocity; (b) Turbulence intensity. Small dashed circles indicate the positions of the fans, while the large circles indicate the rotor-swept area. Measurement points are marked by black dots.

    – Due to the small rotor, the wind generator could be placed so that the rotor operates in the middle of the wind field, so that the high wind field quality could be maintained over the full rotor swept area. Deviations in the rotor position in heave and sway direction of approximately $0.2\,\mathrm{D}$ are tolerable in this context. This limit is by far not exceeded in the present tests.

    – No subdivision of the wind generator's cross-section is created either by modularisation, baffle plates or support structures.

    – A comparably large settling zone is present.

    – Two fine, homogeneous mesh screens without any support structures inside the wind generator cross-section were utilised.

    – An average distance of slightly more than one rotor diameter between the rotor and the wind generator outlet was maintained during the model test.

## 4.2 Model details

A model of the CRUSE Offshore SelfAligner platform (see Figure 2) utilising a single point mooring (SPM) and a downwind rotor was developed and manufactured. A top view of the setup is given in Figure 7. Due to the absence of a yaw mechanism and the ability to turn around the mooring point, the SelfAligner can be considered a passively yawing FOWT concept. The

**Table 1.** Details on selected wind generators utilised in FOWT wave tank tests.

| Test campaign | Meng/SJTU | FOCAL | DeepCWind | Triplespar* | DTU 10 MW TLP* | TUHH/SelfAl. |
|---|---|---|---|---|---|---|
| Size | 3 x 3 m | 7.3 x 3.5 m | 4 x 3 m | 4 x 4 m | 4 x 4 m | 1.6 x 1.6 m |
| D/H[1] | 0.82 | 0.98 | 0.96 | 0.74 | 0.74 | 0.58 |
| Number of fans | 16 | 32 | 35 | 6 | 6 | 16 |
| Number of screens | - | - | - | 4 | 4 | 2 |
| Number of chambers | - | 1 | 1 | 6 | 6 | 1 |
| Guide vanes/baffle plates | yes | - | - | yes | yes | - |
| Honeycombs | - | - | yes (2) | - | - | - |
| Nozzle | - | - | yes | - | - | - |
| Length of settling zone | - | - | > 2 m | unknown | unknown | 1 m |
| Max. turbulence intensity[2] | 13 %[A] | | 7 %[B] | | 15-20 %[D] | <5 % |
| Mean turbulence intensity[3] | | <4.5 % | | | 3.5 %[D] | |
| Max. deviation from avg. speed[4] | 15 %[A] | 25 % | 10 %[B] | 21 %[C] | 14 %[D] | 1.5 % |
| Spatial coefficient of variation [4] | 6 %[A] | 15 % | 6 %[B] | 11 %[C] | 6 %[D] | 1 % |
| Measurement averaging time | unknown | 0.6 s | unknown | unknown | 120 s | 10 s |
| References | Meng (2019) | Lenfest (2023) | Helder (2013) | Bredmose (2017) | Madsen (2020) | this work |

[1] Ratio of rotor diameter to outlet height. [2] Read from graphs/colour maps. [3] Adapted from reference. [4] Own calculations based on values read from graphs/colour maps.

*The same wind generator is used in both campaigns but was characterised individually.

[A] Some measurement points directly at the border of the rotor swept area were excluded as these would impair the results significantly.

[B] The lower tip of the rotor is located at the border of the wind generator. This lowest measurement point is excluded in the analysis as the flow quality is significantly worse at here. In addition, only a limited number of measurement points were available.

[C] Only a limited number of measurement points on a vertical line were available. Measurement series near rated wind was used.

[D] A part of the lower region was excluded from the measurements, which improves the resulting values.

tower is equipped with an airfoil to support the passive yaw mechanism. The airfoil is fully covered by the wind field[2]. A numerical study on the passive yaw capabilities has been presented by the authors in 2020; however, the yaw mechanism is not the focus of this work.

The lower part of the platform (i.e. bottom plate and columns) consists of CNC-milled polyurethane foam with different densities for the bottom plate and columns. Therefore, the geometry and mass distribution of the underwater parts are known exactly, and no deviation due to the soaking of water is expected over time. Inside the hollow columns, fine ballast weights and

330 measurement equipment are stored. The tower consists of an aluminium pipe with carbon fibre bracings connected to the middle columns, which ensure a high tower eigenfrequency far from the wave frequencies. The upper part of the tower subjected

---

[2] Is has to be noted that the flow quality near the lower end of the airfoil around the tower is impaired due to boundary effects of the wind generator

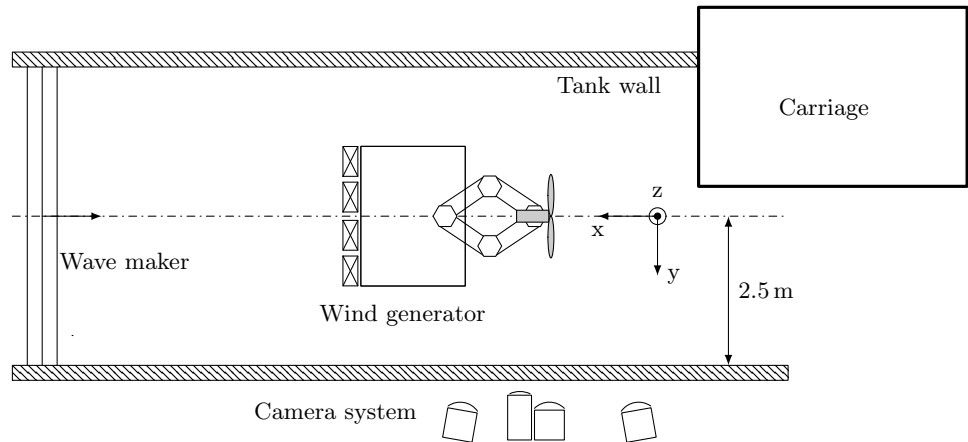

**Figure 7.** Top view of the setup.

to the wind is equipped with an airfoil-shaped cover to maintain a parallel alignment of the platform with the wind.

The two-bladed downwind model wind turbine has been used in wind tunnel investigations focusing on the yaw moment
prior to the wave tank tests. A detailed description of the rotor with a diameter of 0.93 m can therefore be found in Schulz et al. (2022). The same rotor has also been used for an investigation of the unsteady aerodynamic phenomena during a surge motion in the wind tunnel (after the present test campaign), see Schulz et al. (2024). The blades were redesigned for a low Reynolds number regime. However, due to the increased chord length, a comparably high Reynolds number $75x10^3$ at the blades and a power coefficient of approximately 0.35 could be achieved. In order to prevent undesired blade bending, the blades were man-
ufactured using carbon fibre prepreg around a CNC-milled hard resistance foam core with carbon fibre shear webs. A 3D scan of both blades showed negligible manufacturing deviations. A Kollmorgen TBM brushless motor powered by a Kollmorgen AKD controller maintained a constant rotational speed during the tests.

The mooring system had to be redesigned with very short mooring lines due to the limitation of the towing tank width.
Therefore, a similarity to the full-scale system could only be maintained in the static downpull of the mooring system, while the stiffness in the vertical and horizontal directions is different. Commercially available chains (DIN 5685 1/C 30) with a wire diameter of 3 mm kept the scale model in place. However, as no exemplary site was chosen, no specific characteristics of the mooring system had to be fulfilled in the model tests.

The determination of the mass and inertia properties of the wave tank model is described in appendix C. All results are presented with respect to the coordinate system drawn in Figure 7, where the x, y and z axes correspond to surge, sway and heave directions, respectively. Rotations around the x, y and z axis are referred to as roll, pitch and yaw motions.

### 4.3 Sensing system and data acquisition

Motions of the platform in all six degrees of freedom, mooring forces in all three directions, tower top forces and moments in all three directions, rotor speed, wave elevation and wind speed have been continuously monitored during all tests. As reported Robertson et al. (2017) and others (e.g. Ahn and Shin (2019)), the typical cable bundle supplying the measurement equipment and the wind turbine may introduce non-negligible systematic and random uncertainties to the platform motion. Therefore, a wireless data acquisition system was utilised. A detailed description of the sensors and data acquisition system is given in appendix D.

In order to obtain the aerodynamic loads from the tower top measurements, compensation for the inertial and gravitational loads acting on the tower top sensor is necessary. As rotor thrust is of primary interest, the rotor nacelle assembly (RNA) was reduced to a point mass located at the RNA centre of gravity (COG). Then, the instantaneous position of the RNA COG was computed from the measured rigid body position of the model. A simple finite difference method was applied twice, yielding the acceleration of the RNA COG. However, a low pass filter with an edge frequency of 5 Hz was applied to the position signal and the calculated inertia force to reduce noise. This filter frequency is approximately five times higher than the maximum motion frequency and is therefore considered to introduce no systematic error to the rotor thrust. In addition, gravitational loads arising from the inclination of the tower top have also been compensated from the tower top measurements. A hint on the accuracy of the applied compensations can be found in Figure 12, where phase-averaged measurement results (red) of three test cases with waves and without wind are presented. It is shown that the rotor thrust could be compensated to zero with a residual of below 0.2 N for the two cases with higher waves and a slightly higher residual for the case with lower wave height. This corresponds to an expected deviation of approximately 3 % of the mean thrust force during operation.

### 4.4 Load cases

In order to keep the comparison of measurements and simulations as clear and simple as possible, only regular wave cases are presented in this study. A number of wave periods and heights distributed broadly over the expected environmental conditions have been chosen and listed in Table2. Emphasis was put on the cases, including slightly below-rated wind, where a high thrust force occurs and the blade pitch controller is not active.

### 5 Numerical model

The model tests were primarily carried out to provide reliable validation data for the first-order panel method *pan*MARE, which was developed at the Hamburg University of Technology. An extension of the method, which has originally been utilised to calculate the motion behaviour of ship hulls and the loads on ship propellers, to a complete framework for FOWT simulations has been presented by Netzband et al. (2018). In the following, a very brief description of the modelling features is given, while

**Table 2.** Load cases in regular waves.

| Load case | model-scale | | | full-scale | | |
|---|---|---|---|---|---|---|
| | Wave period | Wave height | Wind speed | Wave period | Wave height | Wind speed |
| LCA 1 | 1.19 s | 0.11 m | 0 m/s | 8 s | 5 m | 0 m/s |
| LCA 2 | 1.64 s | 0.18 m | 0 m/s | 11 s | 8.1 m | 0 m/s |
| LCA 3 | 2.09 s | 0.18 m | 0 m/s | 14 s | 8.1 m | 0 m/s |
| LCB 1 | 1.04 s | 0.07 m | 5 m/s | 7 s | 3.1 m | 10 m/s |
| LCB 2 | 1.19 s | 0.11 m | 5 m/s | 8 s | 5 m | 10 m/s |
| LCB 3 | 1.49 s | 0.07 m | 5 m/s | 10 s | 3.1 m | 10 m/s |
| LCB 4 | 1.64 s | 0.18 m | 5 m/s | 11 s | 8.1 m | 10 m/s |
| LCB 5 | 1.79 s | 0.11 m | 5 m/s | 12 s | 5 m | 10 m/s |
| LCB 6 | 2.09 s | 0.18 m | 5 m/s | 14 s | 8.1 m | 10 m/s |
| LCB 7 | 2.30 s | 0.20 m | 5 m/s | 15.4 s | 9 m | 10 m/s |

details can be found in the above-mentioned work by Netzband et al.. In addition, the most relevant modelling parameters are given in appendix E.

*pan*MARE solves hydrodynamic, aerodynamic and mooring-induced loads as well as the motion of the FOWT in time domain. This is a major difference to frequency domain potential flow methods that are often utilised to create inputs for conventional coupled FOWT simulation methods (e.g. for OpenFast with HydroDyn). In this way, the capability of the method to accurately predict fluid loads during aperiodic or large motions and excitations is improved in comparison to conventional FOWT simulation methods. For example, the instantaneous wetted surface of the platform is considered in every time instant, which is not the case in most other FOWT simulation methods. However, this investigation does not consider a deformation of the water surface due to an interaction with the platform. Computing aerodynamic and hydrodynamic flow fields in the same solver allows for a strong coupling of the methods: The 4th-order Runge-Kutta time marching scheme is applied to all sub-models, so that all intermediate steps of the Runge-Kutta scheme are performed simultaneously without a need for additional coupling steps (except for the mooring system).

Pressure forces on the platform hull and the blade surface are determined from the modelled flow fields. However, the un-

derlying potential theory does not account for viscous forces. Therefore, in the hydrodynamic domain, the drag of the platform parts is modelled with drag elements based on empirical coefficients. As the drag coefficients in the model test can vary significantly from the full-scale situation, the coefficients for the simulation model are adjusted to match the decay tests described in the next section.

As described in the previous sections, viscous contributions to the blade loads are dramatically increased in comparison to a full-scale wind turbine. Consequently, the advantage of panel methods to model the pressure distribution around the blades directly becomes less relevant in comparison to the neglection of viscous effects. Therefore, the lifting line sub-module of *pan*MARE is utilised in the aerodynamic domain. The module was originally developed to efficiently generate initial wake geometries for propeller simulations in Wang and Abdel-Maksoud (2020). This sub-module was slightly modified and adopted to wind turbines by the authors. In the sub-module, the discretised blade surface is replaced by a simple lifting line at the 1/4 chord position and the wake is shed from the end of the chord line. In every time step, the local inflow velocity and angle of attack is determined from the flow field, which includes the influence of the wake and all inflow parameters. Then, the lift and drag forces are determined from empirical coefficients, which are determined using the viscous boundary layer solver Xfoil Drela (1989). Finally, the circulation of the lifting line is computed from the lift force.

A lumped-mass mooring model is utilised to account for the mooring line loads. The model considers the axial elasticity while hydrodynamic loads are applied using Morison's equation.

## 6 Uncertainty and data integrity

Due to the complexity of the wave tank tests including a large number of sensors, model characteristics and environmental influences, a precise quantification of the overall uncertainty of the measured quantities is out of scope of this work. Instead, the uncertainty of the most relevant sensors and model characteristics, a number of plausibility checks incorporating different subsystems of the wave tank setup and the results of different repeat tests are presented in the following.

### 6.1 Uncertainties of sensors and model characteristics

In Table 3, the most relevant sources of uncertainties of sensors and model characteristics are given. The sensor uncertainties for the mooring force measurements are quite high in comparison to e.g. the forces observed in surge direction. As no individual calibration protocol is available for this sensor, the absolute uncertainty is computed based on the relative uncertainty given in the data sheet with the maximum load of 100 N. However, this is a very conservative estimation and the accuracy in the measurement region below 20 N, especially when considering the amplitudes instead of the absolute values, is assumed to be much higher.

Another important source of uncertainty is the determination of the rotor averaged wind speed. While the wind speed measurements at one point are comparatively accurate, the non-uniformity of the wind field causes an uncertainty regarding the rotor averaged wind speed because the wind speed at the measurement point may deviate up 1.5 % depending on its position

(see Figure 6a). Therefore, a combined uncertainty of the absolute, rotor averaged wind speed of approximately 2.2 % can be assumed.

As the uncertainty of the motion tracking system is dependent on multiple aspects like the camera positions and the marker positions and geometry, its uncertainty is difficult to quantify. Therefore, a test was performed to estimate this uncertainty. An exactly known weight was placed in the middle of the front floater, while no external force (like mooring system or wind) was acting on the platform. The difference in pitch angle due to the application of the weight was than compared to a simulation with and without the external weight. While the platform pitch angle changed about 3°, the difference between measurement and simulation, which is considered as exact in this simple case, turned out to be 0.027°. As the angle is calculated from the change of positions of the markers in heave and surge direction, an estimation for position uncertainties in these directions can be given based on the distance between the markers (see Table 3). Naturally, this method contains a number of uncertainties. However, the most important sources of uncertainty (mass (< 0.1 %), position of the mass (< 1 % dist to COG) and area moment of inertia of the platform (< 0.5 %)) are considered to be sufficiently low to use this methodology as an estimation for the uncertainty of the motion tracking system.

## 6.2   Plausibility checks

In order to check the level of uncertainty of the combined measurement system, a plausibility check with the aid of simplified simulations was performed. The model was connected to the mooring system and its equilibrium pitch angle, the mooring loads and the thrust force (averaged over 50 s) were measured in wind and no-wind conditions. These loads have been applied as external forces to the simulation model so that again only the static hydrodynamic forces due to the draft of the platform and the resulting pitch angle were calculated within the simulation. The application of the wind loads caused a rotation around the pitch axis of about 2.2°, while the difference between the simulation and measurement was 0.04°. As this is only slightly more than the expected uncertainty of the pitch angle measurement, it is possible that a number of uncertainties canceled out here. However, it seems that the measurement system provides surprisingly accurate results in simple, steady cases.

Another plausibility check is shown in section 7.4, where the residual loads of the inertia removal procedure are shown for three cases without wind loading.

## 6.3   Repeat tests

All tests from Table 2 have been repeated at least once in order to get a hint on the expectable repetition error. For certain load cases, a second repetition was performed. However, these showed very similar deviations and are not shown here for the sake of a clear illustration. The load cases LCB 2, 4 and 6, including short, intermediate and long wave periods, are analysed in detail using phase-averaged data. As the repetition error strongly depends on the wave excitation, the analysis of the repetition error is also performed using these three exemplary load cases. For this and all following analyses, phase averaged data was computed on the basis of measurements, which started after aperiodic effects decayed, which took at least ten motion cycles after the approaching of the first wave. The data sets had a length of 6 - 12 motion cycles, depending on the quality of the generated waves and the occurrences of obstacles or noise in certain measurement channels. In Figure 8, wave elevations,

**Table 3.** Uncertainties of sensors and model characteristics.

| Quantity | Uncertainty | Source |
|---|---|---|
| Tower top force | 0.128 N / approx. 2 % of rotor thrust | ME-Systeme K6D calibration protocol |
| Mooring force surge | 0.5 N | Althen ALF233 data sheet |
| Mooring force heave | 0.5 N | Althen ALF233 data sheet |
| Motion tracking system pitch angle | approx. 0.03° | estimation (see 6.1) |
| Motion tracking system heave/surge position | < 0.1 mm | estimation (see 6.1) |
| Wind speed measurement | 0.07 m/s | derived from data sheet of difference pressure sensor |
| Rotor averaged wind speed | 2.2 % | derived form sensor uncertainty and wind field non-uniformity |
| Model weight measurement | < 0.1 % | data sheet |
| Platform geometry | | |
| - Distances 0.02 - 0.2 m | < 1 % | manufacturing tolerance |
| - Distances > 0.2 m | < 0.25 % | manufacturing tolerance |
| - Water plane area | < 0.5 % | manufacturing tolerance |
| Rotor geometry | see Schulz et al. (2022) | |

selected platform motions and selected force measurements recorded for each load case and two test runs each are plotted.
The red and orange lines illustrate the phase-averaged quantities computed from a minimum of five motion cycles for both test runs. Multiple periods of the time domain signals are indicated as black and grey dots. All loads and motions are shown as absolute values in model-scale in order to allow for the evaluation of relations between mooring, tower top loads and surge motion directly. This illustration is also used in the following plots containing phase-averaged data.

The phase-averaged wave elevation shows nearly no deviation between the two test runs. However, in 8a, a slightly increased scattering of the time signal is noticeable. Concurrently, the wave shape does not follow the intended sine function, which indicates that the wave maker is driven near its limitations. Interestingly, the same wave input produced a different wave shape during the wave-only load cases, which are shown in Figure 12a. However, the repetition of this case showed the exact same wave shape. Therefore, it is likely that the internal configuration of the wave maker was slightly changed between the two. This effect seems to be limited to this load case, as the other wave configurations could be reproduced in a satisfactory manner. In 8a, a high scattering of the surge motion in comparison to its amplitude can be noticed, which is due to a slow drift motion arising from the start of the wave maker. However, the surge motions are very small in this case. The remaining platform motions could be reproduced with a very low repetition error and an increased scattering when considering the lowest wave period. In the illustrations of the aerodynamic rotor thrust, a significantly stronger scattering can be seen. This arises from the superposition of aerodynamic, inertia and gravitational forces in the measurement of the tower top force in wind direction. The random uncertainty occurring in the measurement of the significantly higher inertia loads cannot be compensated during the calculation of the rotor thrust force and, therefore, adds considerable scattering to it. In the wave-only cases in Figure 12, the

residual rotor thrust force is shown. However, the observed scattering in the repetition tests is slightly higher, which leads to the conclusion that the aerodynamic thrust also adds a part to the random uncertainty. At the lowest wave period, the compensation

of inertial and gravitational forces seems to be unable to properly reproduce the variation of the thrust force due to the tower top motion, while a satisfactory repetition error could be reached in the other load cases. Therefore, the calculated rotor thrust has to be considered carefully at low platform excitation. The vertical mooring force could also be reproduced with a low repetition error. In summary, the absolute phase-averaged repetition error and scattering do not vary strongly from case to case, which yields stronger relative deviations for cases with less prominent platform motions.

## 490 7 Results

The comparison and analysis of the experimental and numerical results is divided into five sections. First, a comparison of different rotor thrust force measurements and corresponding simulations is presented. Second, the damping behaviour and eigenfrequencies of the platform are evaluated in decay tests. Third, phase-averaged results of hybrid simulations and the corresponding measurements are presented to separately examine the contribution of inaccuracies arising from the mooring and

495 aerodynamic simulation modules in three cases. Next, the differences between full simulations and experiments in the same three cases are examined with and without wind during one phase-averaged motion cycle. Finally, the normalised motion amplitudes of all cases with wind are examined in order to examine the influence of the motion frequency on the inaccuracies of the simulation model.

### 7.1 Simulations and measurements of the rotor thrust

In Figure 9, thrust force measurements of a testing campaign performed in the wind tunnel of TUHH (dots) are shown together with numerical simulations performed with *pan*MARE (lines) at different tip speed ratios (TSR). In addition, a black cross indicates the thrust coefficient measured in the wave tank setup. The wind tunnel measurements have been performed at two

different rotor speeds (750 and 1050 RPM), which corresponds to Reynolds numbers of approximately $95x10^3$ and $125x10^3$ at the middle and outer parts of the blade. As the lift coefficient of the utilised SD7062 airfoil is insensitive against changes in the Reynolds number in this region, no significant differences can be observed between the measurement series. The lifting line simulations have been performed with lift and drag coefficients, which where individually calculated with Xfoil at the corresponding Reynolds numbers. A nearly precise agreement with the measurements in the lower TSR range and and a slight

overprediction at high TSR can be concluded from the comparison. The influence of the Reynolds number is a very slight offset, which is caused by slightly lower lift coefficients predicted by Xfoil for the lower Reynolds number. Measurements in the wave tank setup were performed at a constant TSR of 6 and a blade section-based Reynolds number of approximately $75x10^3$. The average thrust coefficient was measured over a period of 10 s at still water conditions with a platform pitch angle of -0.66°. The deviations of the tower top surge position from the mean were below 0.02 m and resulting tower top velocities

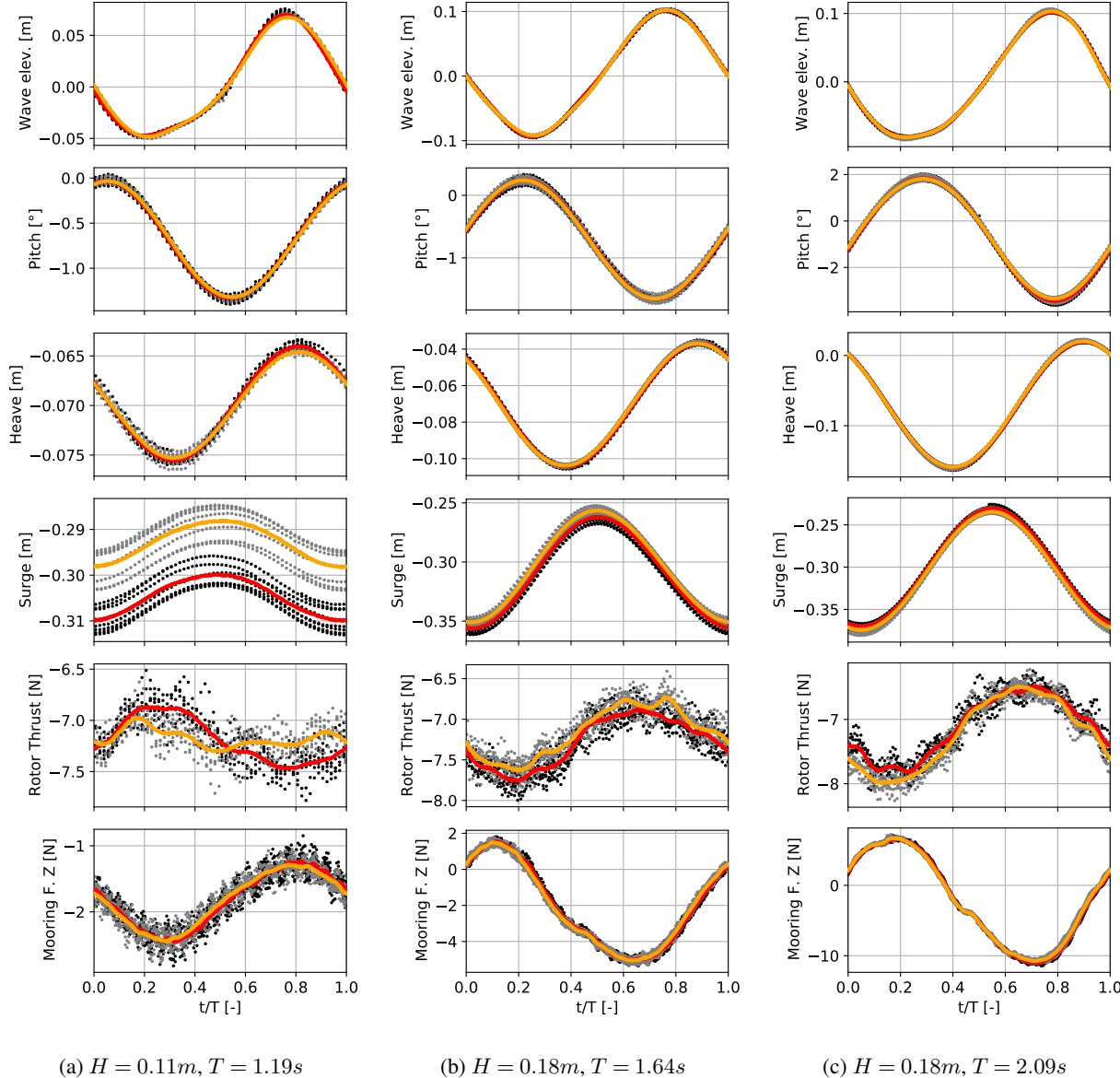

(a) $H = 0.11m$, $T = 1.19s$    (b) $H = 0.18m$, $T = 1.64s$    (c) $H = 0.18m$, $T = 2.09s$

**Figure 8.** Waves and wind: Repetition tests at three different wave conditions (wave height $H$ and period $T$). Experiment 1st run: ········ and ───── (averaged); Experiment 2nd run: ········ and ───── (averaged).

below 0.05 m/s. Therefore, the setup can be considered as comparable to the wind tunnel setup although the platform was a free-floating state. Present Xfoil predictions and experimental investigations in literature (see Lyon et al. (1997)) show a consistent trend of a slightly decreasing lift coefficients with lower Reynolds numbers for the utilised airfoil in the considered operation region. Therefore, a slight decrease of the thrust coefficient at a given TSR would be expected for the measurement in the wave tank. In contrast to this, a slight increase of the thrust coefficient in comparison to the wind tunnel study is visible,

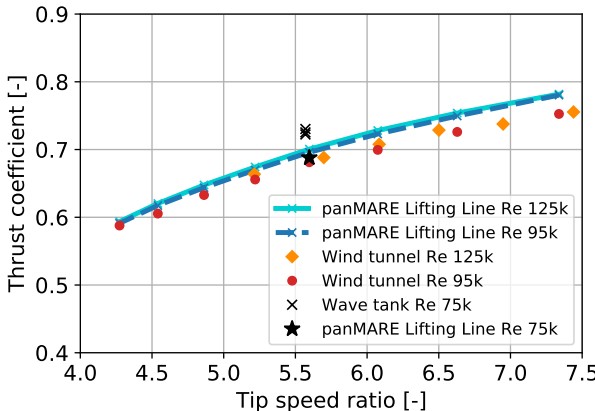

**Figure 9.** Thrust coefficient of the TUHH model wind turbine over varying TSR in wind tunnel and wave tank wave tank measurement campaigns as well as in simulations. The TSR variation is achieved by a variation of the wind speed and a constant rotational speed so that the blade section-based Reynolds number is approximately constant for a measurement series.

while the simulations show a minimal decrease at a Reynolds number of $75x10^3$ (black star). This is most likely caused by the uncertainty of the of the mean wind speed measurement in the wave tank setup, which is discussed in 6.1. Due to an underprediction of the absolute wind speed in a range of 2.2 %, an increase of the thrust coefficient of nearly 5 % would result. As such underprediction is within the range of uncertainty of the absolute, rotor averaged wind speed as discussed in 6.1, this and the sensor uncertainty itself (approx. 2 %) are considered as the reason for the consistently higher thrust force in the wave tank setup. In contrast to this, the a low repetition error can be concluded from the three repetitions shown in Figure 9. The maximum deviation from the mean value is approx. 0.6 % of the rotor thrust force, which gives a strong hint that the wind loads do not increase the repetition error of the platform motions significantly.

## 7.2 Decay tests

Decay tests in heave, surge and pitch were performed. In the case of heave decay, the mooring system was not connected to the floater in order to identify the heave damping separately from the pitch motion, which would have been introduced by the mooring system. From the decay tests in heave and pitch in Figure 10a and 10b, it can be concluded that eigenfrequencies, as well as motion damping in both directions, can be predicted accurately in the simulations. In the surge direction, a slightly larger difference in the eigenfrequency in the simulation and experiment was observed, while the damping could be precisely met. A slight mismatch of the eigenfrequency indicates that the mooring stiffness in the surge direction is slightly overpredicted. From a comparison of the mooring loads in surge and heave direction, it is obvious that a strong pitch-surge coupling is present. This coupling cannot be exactly reproduced in the simulation. Although not directly visible in these results, a pitch-heave coupling is evident from the excentric application point of the SPM. Therefore, a coupling of heave, pitch and surge motion is likely

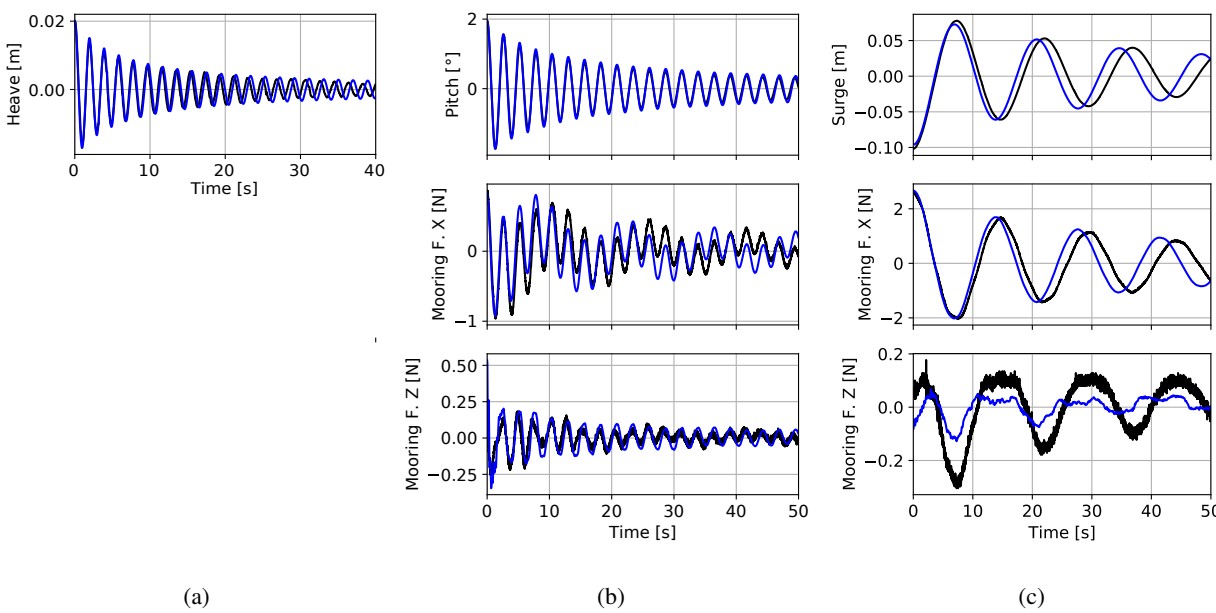

**Figure 10.** heave (a), surge (b) and pitch (c) decay tests. Experiment: ———— ; Simulation: ———— .

to strongly influence the motion behaviour of the considered FOWT. During the surge decay, the mooring force in the heave direction is not met properly, which is due to the extremely low motion amplitude.

### 7.3 Hybrid simulations

In Figure 11, measurement results and simulated forces from hybrid simulations are shown. In these cases, the simulation model synchronously undergoes the exact same rigid body motions as the experiment. In the upper two rows of the picture,

the compensated rotor thrust force and the originally measured tower top force in the surge direction are shown. A consistent underprediction of the average rotor thrust of approximately 5 % is apparent for all three wave periods, which is in line with observations under steady conditions. This is also the case in the tower top force. However, it is barely visible due to the oscillation amplitude, which is nearly one order of magnitude higher compared to the rotor thrust. From the ratio of the tower top force and the rotor thrust amplitudes, it becomes obvious that comparably small uncertainties in the tower top force may

fundamentally impair the quality of the rotor thrust measurement. However, the amplitudes of rotor thrust force and tower top force seem to be captured well in all three cases. The meaningfulness of the good agreement of the rotor thrust force in 11a may be doubted because the repetition of the measurement showed different behaviour in Figure 8a.

Stronger deviations between measurements and simulations were observed in the mooring loads. An atypical, high-frequent behaviour can be seen in the measured forces, especially in the horizontal direction. The simulations show a similar dynamic

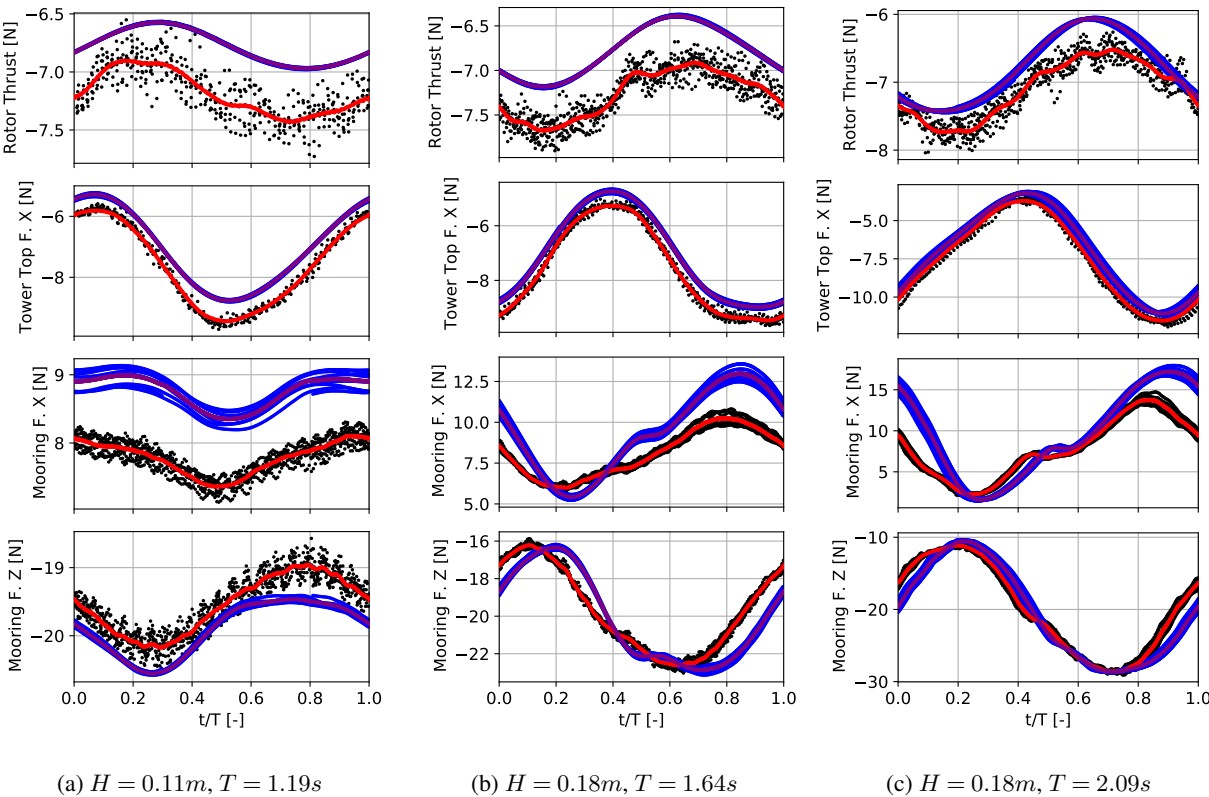

(a) $H = 0.11m,\ T = 1.19s$     (b) $H = 0.18m,\ T = 1.64s$     (c) $H = 0.18m,\ T = 2.09s$

**Figure 11.** Waves and wind: Hybrid simulations and experimental results at three different wave conditions. Experiment: ········ and ——— (averaged); Simulation: ········ and ——— (averaged).

behaviour. However, strong differences in the minimum and maximum force in the surge direction are present. In addition, the previously mentioned slow surge motion at the lowest wave period introduced a large scatter in the phase-averaged diagram. In the heave direction, an acceptable match of the minimum and maximum force could be achieved.

In summary, an atypical behaviour of the mooring system and a poor match of simulations and measurements in surge direction can be stated for the mooring loads. It is likely that the extremely short mooring lines due to the limited space in the towing tank caused this behaviour, which is obviously beyond the limitations of the lumped mass mooring method.

## 7.4  Full simulations

Measured and simulated motions and loads of LCA 1, 2 and 3 in waves without wind (corresponding to LCB 2, 4 and 6 with wind) are shown in Figure 12. The obvious distortion of the wave shape in LCA 1 has already been discussed in the repeat tests section. The simulated waves were abstracted from the measured wave elevation using an inverse Fourier transformation and applied at the exact position of the wave sensor in the simulation. The accuracy of this method has not been investigated in

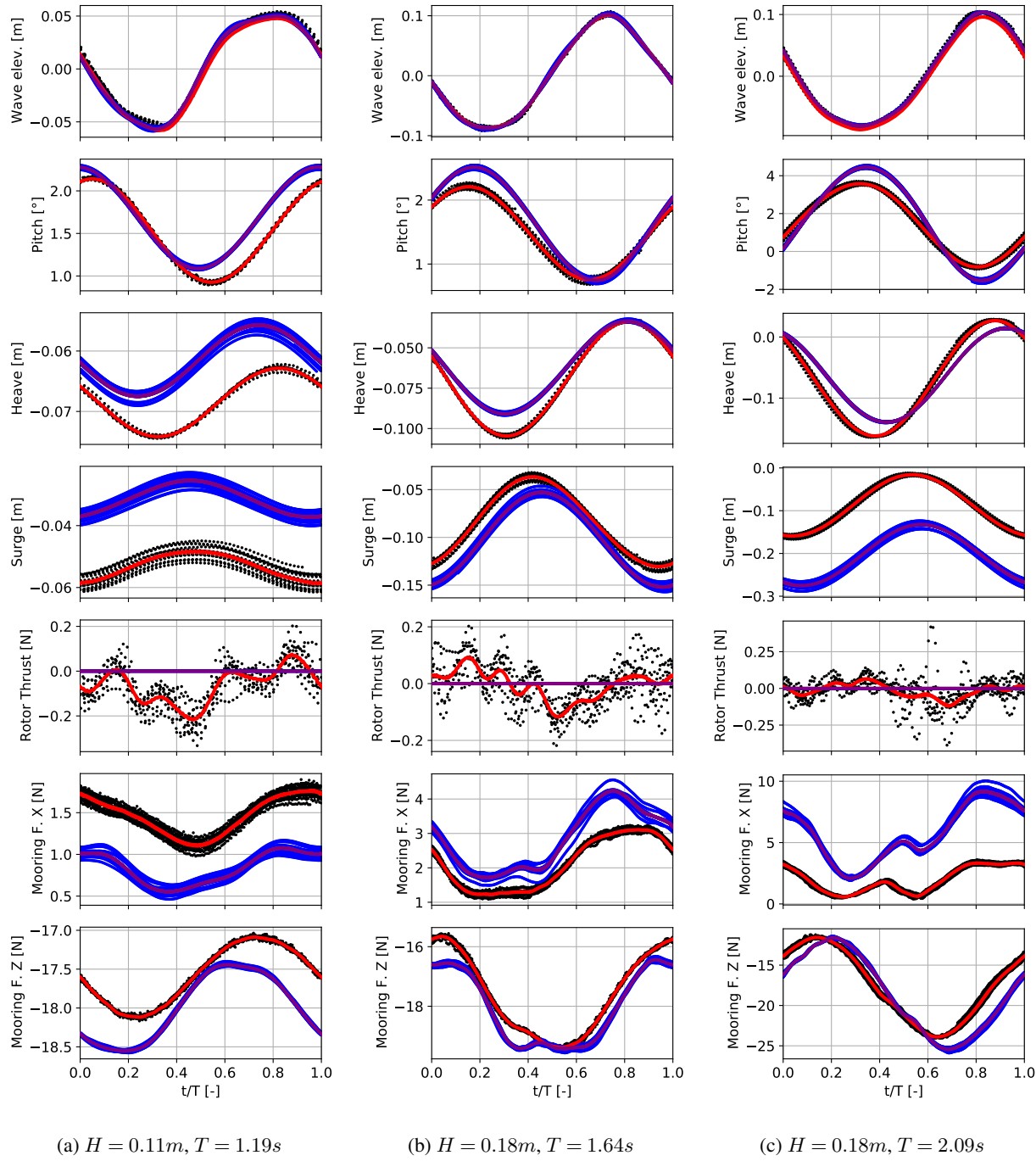

(a) $H = 0.11m, T = 1.19s$

(b) $H = 0.18m, T = 1.64s$

(c) $H = 0.18m, T = 2.09s$

**Figure 12.** Waves only: Full simulations and experimental results at three different wave conditions. Experiment: ⋯⋯⋯ and ▬▬▬ (averaged); Simulation: ⋯⋯⋯ and ▬▬▬ (averaged).

detail, which implies that only a rough comparison of the phases between measured and simulated motions can be performed. Furthermore, the comparison of the wave elevations reveals that the waves applied in the numerical method are very similar, but do not exactly match the measured surface elevation. A satisfactory match between simulations and measurements of the motions in shape, amplitude and phase is achieved, whereas the pitch amplitude tends to be overestimated, and the heave amplitude tends to be underestimated at higher motion periods. This indicates that the heave-pitch (or heave-pitch-surge) coupling is not exactly captured in the simulations. As the single point mooring is a major contributor to this coupling, it is likely that the mismatch of the mooring loads, revealed by the hybrid simulations, is responsible for this issue. With a rising wave period, the simulated mean surge drift rises stronger than observed in the experiments. This offset in the surge direction can be explained by the mean mooring force in the surge direction, which makes clear that the numerical model is unable to accurately predict the mean wave drift forces. A possible source of this modelling inaccuracy is the neglect of the deformation of the water surface induced by the platform, which may lead to a wrong contribution of the pressure forces to the surge force. While the mooring forces in the vertical direction are met with an acceptable error for LCA 2 and 3, a considerable relative deviation between the maximum values is present in LCA 1. In addition to the general modelling issues regarding the mooring system, the difference in the mean surge position may also play a role here. Apart from the surge mean value, differences in a similar magnitude between experiment and simulation can be found in the mooring loads when considering the full and the hybrid simulations. Therefore, it is evident that a considerable part of the modelling inaccuracies observed may be ascribed to the scaling issues of the mooring rather than to issues in the hydrodynamic simulation or coupling.

The same wave conditions but including a wind field with a velocity slightly below the rated wind speed of the turbine were applied in LCB 2, 4 and 6, which are shown in Figure 13. When considering the time domain signals indicated with black dots, the applied wind field caused an increase of absolute scattering in the mooring forces compared to the wave-only cases. Nevertheless, no significant changes in the scattering can be seen in the motions. Overall, an agreement of simulated and measured platform motions comparable to the wave-only cases could be achieved, as the pitch and heave amplitudes are again slightly over- i.e. underestimated. The mean value and fluctuation of the rotor thrust force due to the tower top motion are clearly visible in the measurements and cause an offset of the mean surge position and pitch angle in all cases. In addition, the minimum aerodynamic thrust occurs during the forward surge and pitch motion, which clearly indicates the presence of aerodynamic damping. However, when comparing the measured and simulated thrust force amplitude, an overestimation, rising with the wave period, is apparent. As this clear overestimation is not present in the hybrid simulations, it is most likely that this mismatch is caused by the higher simulated amplitude of the tower top motion in the surge direction due to the platform pitch and surge motions.

Compared to the wave-only cases, an increase in the pitch motion amplitude due to the presence of wind can be noticed. This contradicts the above-mentioned observations and experiences from other experiments, where the aerodynamic damping reduced the platform pitch amplitude. The influence of the mean thrust force on the mooring system may provide an explanation for this. While the platform undergoes similar surge motions in with and without-wind cases, the mooring force amplitude in the surge direction is more than doubled. It can, therefore, be concluded that the mooring stiffness has increased drastically due

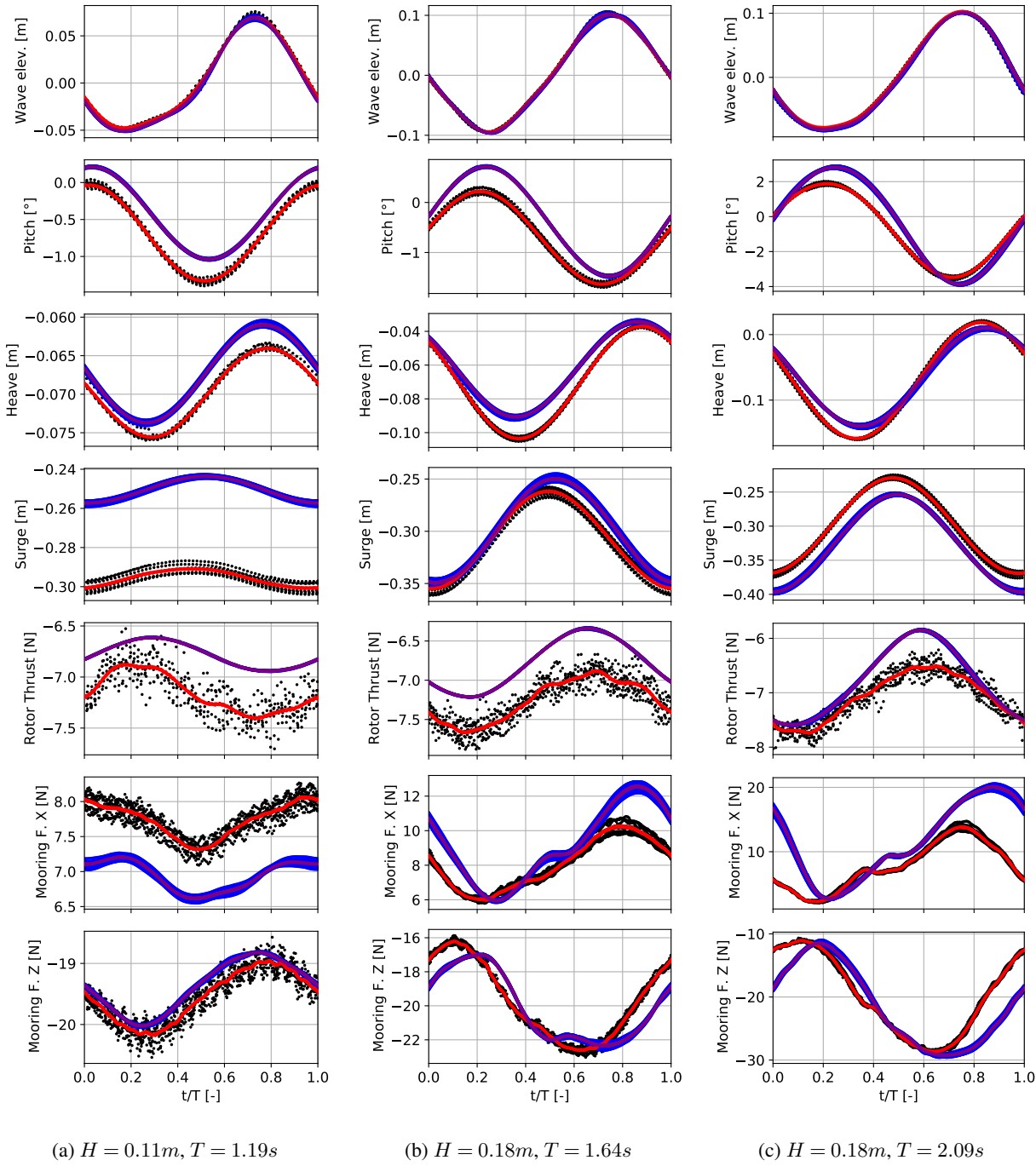

(a) $H = 0.11m, T = 1.19s$      (b) $H = 0.18m, T = 1.64s$      (c) $H = 0.18m, T = 2.09s$

**Figure 13.** Waves and wind: Full simulations and experimental results at three different wave conditions. Experiment: ········ and ▬▬▬ (averaged); Simulation: ········ and ▬▬▬ (averaged).

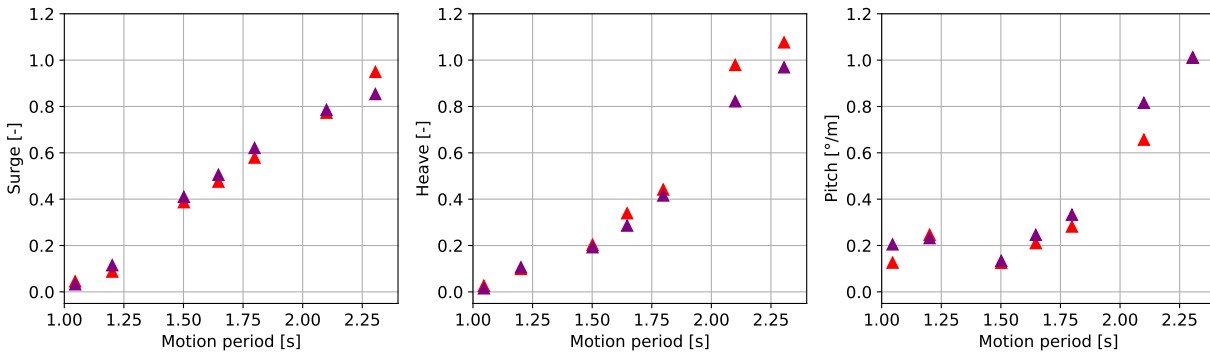

**Figure 14.** Waves and wind: Normalised motion amplitudes of regular wave and wind cases. Experiment: ▲, Simulation: ▲

to the mean surge deflection. As a consequence, the effect of heave-pitch-surge coupling induced by the single point mooring is increased, which finally overcompensates the aerodynamic damping and leads to a stronger pitch motion. As discussed in the previous sections, the mooring system could not be scaled properly, such that the stiffness in the surge direction is most likely stronger than in reality. In addition, the influence of aerodynamic damping is reduced due to the downscaling of the rotor. It is, therefore, unlikely that the aerodynamic damping is overcompensated in such a way in full-scale. However, the increased influence of heave-pitch-surge coupling on the platform pitch due to the presence of wind should also be considered in full-scale. From the comparison, it could be demonstrated that the simulation is generally able to capture the effect of a mean surge deflection on the heave-pitch-surge coupling but suffers from deviations in the absolute mooring forces.

### 7.5 Motion amplitudes

In Figure 14, the motion amplitudes obtained from load cases LCB 1-7 are shown normalised by the wave elevation. An increase in the motion response is visible in heave and pitch with a rising wave period. The identified issues in reproducing the mooring loads leading to a deviated contribution of the heave-pitch-surge coupling to the platform motion can also be seen in this illustration. Nevertheless, an overall good agreement between measurements and simulations with slightly higher deviations at long wave periods can be concluded from the comparison of the normalised amplitudes.

### 8 Conclusions

In this study, the application of an improved testing and validation strategy for FOWT experiments and simulations is presented. Model tests of the CRUSE Offshore SelfAligner concept were performed in a wave basin, and the first-order panel method *pan*MARE and its lifting line sub-module are validated on the basis of the measurement data. In order to allow for a compensation of inertia and gravitational forces from the tower top loads revealing the aerodynamic rotor thrust force, a significant increase of the flow quality in comparison to conventional mobile wind generators was needed. This was achieved

by an elaborate wind generator design, which would have been too large to fit the wave basin. Therefore, a reduction of the size of the wind turbine and thus the wind generator by a factor of three in comparison to common Froude scaling was applied. The consequences of the violation of scaling rules are discussed and found to be acceptable in exchange for a higher flow quality when aiming at the validation of a numerical method. Aside from the fully coupled simulations, hybrid simulations were performed to improve the distinction between causes and effects regarding differences between measurements and simulations, which is a general challenge in FOWT validations.

The major findings from the tests and the analysis are grouped in three sections:

**Uncertainties**

- The analysis of the expectable accuracy of model characteristics, environment, sensors and post processing showed that the uncertainties have negligible influence on the comparison between simulation and experiment in most cases. Especially the non-uniformity of the wind field could be decreased by a factor of approximately five compared to other experiments, which yields an expected systematic wind speed error of slightly more than 2% of the nominal wind speed. This cloud be successfully verified in a comparison with a wind tunnel measurements of the same rotor.

- The repetition error of the platform motion could be reduced to a level slightly higher than the repetition error of the waves. Thus, only a marginal amount of random uncertainty was added to the measurements by the aerodynamic loading, which is a consequence of the low repetition error of the thrust force of approximately 0.6 % (in steady conditions).

- The inertia removal procedure for the calculation of the aerodynamic rotor thrust from the tower top force could be validated and showed a maximum uncertainty of 3 % of the nominal thrust force.

**Validation of *pan*MARE**

- A mismatch of the mooring loads was the major driver of deviations between measurements and simulations. This was caused by an insufficient scaling of the catenary single point mooring due to the limited width of the towing tank, which lead to a highly dynamic behaviour.

- A strong coupling of motions in heave, pitch and surge direction was introduced by the single point mooring. This coupling is characteristic for single point mooring structures.

- Due to the influence of the single point mooring system applied in the wave tank, a negative aerodynamic damping of the platform pitch motion was observed. It is not entirely clear whether this effect will appear similarly in full-scale.

- The simulated and measured aerodynamic loads showed very good agreement in time domain without a calibration of the simulation model, which is a novelty for comparisons of simulations with wave tank experiments of FOWT.

- Despite the scaling issue of the mooring system, an overall good agreement between measurements and fully coupled simulations could be achieved.

**Validation strategy**

- Performing hybrid simulations based on the measured platform motions together with the measurement of all relevant
external loads turned out to be of major value because it allowed for the use of the exclusion principle. As a consequence,
the sources of inaccuracies between measurements and simulations could be clearly identified in different load cases.

- Comparisons of all measured and simulated quantities using phase averaging allowed for the conservation of all relevant
information from the measurements while random uncertainty was reduced to an extremely low level for most measured
quantities.

- The reduction of the rotor size, which was mandatory to achieve an improved flow quality, resulted in a significantly
lower contribution of aerodynamic damping to the platform pitch motion. However, this contribution was well-defined
and could be reproduced accurately in the simulations.

Finally, the proposed improvements to the validation process proved to be valuable in the presented case. Especially the use
of a smaller scaling ratio for the rotor than for the rest of the structure offers the chance to easily allow for an improved wind
field quality. In the present case, this offered further opportunities to improve the quality of the validation, such as the accurate
measurement of the rotor thrust force. Nevertheless, the reduced sensitivity of the rotor thrust to tower top motions violates the
Froude similarity and needs to be evaluated carefully for the desired application case.

*Data availability.* The raw data of the simulation results and measurements can be provided by contacting the corresponding author. The
platform design is proprietary. However, a use of the design for research purposes is desired and can be discussed on request.

**Appendix A:  Rotor scaling**

In this section, the influence of the reduced rotor size in comparison to conventional Froude scaling is discussed. As described
in section 3, the geometric scaling factor for the rotor diameter ($\lambda_{aero}$) is not identical with the one corresponding to the rest
of the platform ($\lambda_{hydro}$). As it is the aim of the scaling procedure to keep the mean thrust force of the further downscaled rotor
($T_{present}$) consistent with the thrust force that would arise from a the application of conventional Froude scaling ($T_{\lambda_{hydro}}$),
the wind speed needs to be increased so achieve the same mean thrust force with a smaller rotor. The mean thrust force of the
ideally Froude scaled rotor and the present model rotor can be calculated as follows:

$$T_{\lambda_{hydro}} = \frac{1}{2} \rho A_{\lambda_{hydro}} C_t u^2_{\lambda_{hydro}} \tag{A1}$$

$$T_{present} = \frac{1}{2} \rho A_{\lambda_{aero}} C_t u^2_{present} \tag{A2}$$

where $\rho$ and $C_t$ denote the air density and nominal thrust coefficient. The rotor swept area of the ideally Froude scaled rotor and
the further downscaled rotor are described as $A_{\lambda_{hydro}}$ and $A_{\lambda_{aero}}$, respectively. The wind speed according to Froude scaling

is $u_{\lambda_{hydro}}$, while $u_{present}$ stands for the wind speed applied to the further downscaled rotor in the proposed setup. In order to achieve the targeted mean thrust force for an ideally Froude scaled model system ($T_{\lambda_{hydro}}$) using the present rotor with reduced size, the wind speed need to be higher compared to conventional Froude scaling. This can be derived as follows:

$$T_{\lambda_{aero}} \overset{!}{=} T_{\lambda_{hydro}}. \tag{A3}$$

$$\frac{1}{2}\rho A_{\lambda_{aero}} C_t u_{present}^2 = \frac{1}{2}\rho A_{\lambda_{hydro}} C_t u_{\lambda_{hydro}}^2 \tag{A4}$$

In an intermediate step, the rotor swept area can be expressed in terms of the full-scale rotor radius $R_{fs}$.

$$A_{\lambda_{aero}} = 2\pi \left(\frac{R_{fs}}{\lambda_{aero}}\right)^2, A_{\lambda_{hydro}} = 2\pi \left(\frac{R_{fs}}{\lambda_{hydro}}\right)^2 \tag{A5}$$

Substituting this into equation A4 and solving for the ratio of $u_{present}$ and $u_{\lambda_{hydro}}$ yields:

$$\frac{u_{present}}{u_{\lambda_{hydro}}} = \frac{\lambda_{aero}}{\lambda_{hydro}}. \tag{A6}$$

Therefore, the present reduced model rotor delivers the same mean rotor thrust as an ideally Froude scaled rotor, when the wind speed is increased by a factor of $\lambda_{aero}/\lambda_{hydro}$ in comparison to the Froude scaled wind speed. However, the above derivation only covers the mean wind rotor thrust. The consequences of the present scaling approach in terms of the variation
of the thrust force can be approximated easily when considering the following simplifications: The tower top motion velocity is low in comparison to the wind speed. The thrust coefficient is approximately constant for small changes in the effective wind speed seen by the rotor and independent of the derivative of the effective wind speed during a motion cycle. With these simplifications, the thrust force fluctuation amplitude due to a sinusoidal tower top motion in the surge direction can be described as follows.

$$T_{amp,\lambda_{hydro}} = \frac{1}{4}\rho A_{\lambda_{hydro}} C_t \left( \left(u_{\lambda_{hydro}} + v_{tt,max}\right)^2 - \left(u_{\lambda_{hydro}} - v_{tt,max}\right)^2 \right) \tag{A7}$$

$$T_{amp,present} = \frac{1}{4}\rho A_{\lambda_{aero}} C_t \left( \left(u_{present} + v_{tt,max}\right)^2 - \left(u_{present} - v_{tt,max}\right)^2 \right) \tag{A8}$$

In equation A7 and A8, the thrust force fluctuation amplitude is approximated using Froude scaling ($T_{amp,\lambda_{hydro}}$) and the
705 proposed further downscaling of the rotor ($T_{amp,present}$). $T_{amp,\lambda_{hydro}}$ is considered as a reference value reflecting the full-scale behaviour. $v_{tt,max}$ denotes the maximum tower top surge velocity. As mentioned in section 3, it is mandatory to ensure that the absolute value of the mean thrust $T_{\lambda_{hydro}}$ and $T_{\lambda_{aero}}$ remains the same during subscaling and at the same time TSR is kept constant. Therefore, the thrust coefficient remains unchanged. In other words, due to an increase in the inflow speed and

the rotational speed, the absolute value of the thrust of the smaller rotor increases to the desired level while the operating point
(TSR) is kept the same.

The ratio of the conventionally scaled and the further downscaled rotor thrust force fluctuation amplitude can be simplified
to:

$$\frac{T_{amp,present}}{T_{amp,\lambda_{hydro}}} = \frac{A_{\lambda_{aero}} u_{present}}{A_{\lambda_{hydro}} u_{\lambda_{hydro}}} \tag{A9}$$

As $A_{\lambda_{aero}}$ can be derived based on the increased scaling factor $\lambda_{aero}$ and $u_{present}$ is determined by the ratio of the aerodynamic
and hydrodynamic scaling factors (see equation A6), equation A9 can be simplified using the expressions in equation A10.

$$A_{\lambda_{aero}} = A_{\lambda_{hydro}} \left( \frac{\lambda_{hydro}}{\lambda_{aero}} \right)^2, u_{present} = u_{\lambda_{hydro}} \left( \frac{\lambda_{aero}}{\lambda_{hydro}} \right) \tag{A10}$$

$$\frac{T_{amp,present}}{T_{amp,\lambda_{hydro}}} = \frac{\lambda_{hydro}}{\lambda_{aero}} \tag{A11}$$

From equation A11, it is evident that the proposed approach leads to a reduced thrust force amplitude by a factor of
$\lambda_{hydro}/\lambda_{aero}$.

## Appendix B: Determination of flow quality measures

In Table 1, a number of flow quality measures are displayed for wind generators used in previous wave tank tests. Due to the
lack of published data, the flow quantities are calculated from graphs or color maps from the corresponding references. In
case of color maps, the quantity (mean wind speed or turbulence intensity) is evaluated on 12 points on a straight line through
the rotor swept area, which is placed so that the highest deviations are covered. The evaluation of the values is therefore
not accurate, but an estimation. For the calculation of the spatial coefficient of variation of the wind speed ($CV(u_{line})$), the
following equation is used:

$$CV(u_{line}) = \frac{std(u_{line})}{\overline{u_{line}}}, \tag{B1}$$

where $std(u_{line})$ and $\overline{u_{line}}$ denote the standard deviaton and mean value of the velocities evaluated on the line.

It has to be noted that the operation conditions and measurement setups in the different references are not identical, which leads
to a limited comparability of these values. One of the most important differences might be the averaging time for the calculation
of the mean wind speed, which deviates from 0.6 s to 120 s and is unknown for some cases. The short averaging time most
likely impairs the quality measures for the wind generator used in the FOCAL test campaign. In addition, the calculation of the
turbulence intensity may deviate from reference to reference. Therefore, it is important to keep in mind that the flow quality
measures can only give an indication for the comparison of the wind generators.

## Appendix C: Determination of mass and inertia properties

A considerable effort was invested to determine the weight and exact position of every part of the model, which includes all screws and cables. All measured weights and positions were fed into the CAD model in order to minimise the differences between the 3D model and the real model. Finally, the mass moments of inertia and exact COG position have been extracted from the CAD model and applied to the simulation model. In order to give a hint on the reliability of this methodology, the model was placed on three pins with predefined positions, which were marked during the CNC manufacturing. Then, three scales were placed below the pins so that the COG position in the lateral plane could be computed from the weight measurements and the predefined distances. The difference between the lateral COG positions obtained in this way and from the CAD model turned out to be 1.4 mm and 2.2 mm respectively. Normalised to the platform length and width, this corresponds to a relative deviation of 0.07 % and 0.18 %. Therefore, a low systematic uncertainty arising from differences in the mass moment of inertia and COG position is expected.

## Appendix D: Sensing and data acquisition systems

### D1 Sensing systems

A Vicon optical motion tracing system was utilised to capture the platform motion. The motion of a specialised marker geometry consisting of reflecting spheres is computed from the pictures of three cameras positioned outside the towing tank by the processing unit. The horizontal alignment of the measured platform inclination was calibrated with a setup where the platform was aligned with the water surface. This was achieved by ballasting the platform so that all columns had the same draft, which was verified by an ultra-sonic-based distance sensor mounted above the floater model. The mooring forces have been measured using the three-component force sensor Althen ALF233, which fulfils the IP67 standard. A mould was milled on the lower surface of the bottom plate during manufacturing to accommodate the sensor so that the shape of the underwater geometry would not be altered when the sensor was mounted. Proper alignment in the yaw direction was reached with the aid of pin holes in the sensor and bottom plate. A me-systeme K6D six-component force/moment sensor was mounted directly below the nacelle in order to capture all relevant aerodynamic, inertial and gravitational loads. An ultra-sonic-based surface elevation sensor was placed at a defined position relative to the platform equilibrium position, but it was near the side of the tank in order to reduce the influence of wave reflections from the model in the measurements. The information on the exact sensor placement allowed for a precise application of the waves in the later simulations without the need for a phase shift correction. Hall sensors integrated into the Kollmorgen motor were utilised to measure the rotor speed of the wind turbine.

### D2 Data acquisition systems

Two bluetooh-based wireless measurement amplifiers, me-systeme GSV6BT and GSV3BT, were utilised to transmit the tower top loads and mooing forces to the data acquisition computers. While the GSV3BT was placed inside the front column, the GSV6BT could be integrated into the tower top due to its low overall weight of approximately 100 g. The wave elevation sensor

**Table E1.** Most relevant simulation parameters.

| Model parameter | Value | Comment |
| --- | --- | --- |
| *Platform* | | |
| - Number of panels | 2776 | |
| - Number of drag elements | 60 | |
| - Drag coeff.: bottom element vertical | 2.25 | |
| - Drag coeff.: bottom element horizontal | 2.25 | |
| - Drag coeff.: floater element horizontal | 0.642 | |
| *Waves* | | |
| - Number of Fourier frequencies | 400 | used to reproduce the measured wave elevation |
| *Mooring* | | |
| - Number of chain elements p.l. | 30 | |
| - Diameter | 0.00353 m | |
| - Mass in water | 0.122 kg/m | |
| - Displaced volume | 0.0000173 $m^3/m$ | |
| - Drag coefficient traverse motion | 1.15 | |
| - Stiffness (EA) | 1200 | |
| *Rotor* | | |
| - Number of lifting line elements p. bl. | 20 | refinement at tip and root |
| - Number of wake panels | 8.000 | corresponds to a wake length of more than 5 rotor diameters |
| - Number of freely deforming wake panels | 1.280 | |
| - Desingularisation radius | 0.088 m | |
| *Tower* | | |
| - Number of lift and drag elements | 10 | only modelled in the wind field |

was operated with a land-based measurement amplifier. All measurement amplifiers, as well as the motion tracking system, were synchronised using a rectangular pulse triggered by a Wireless LAN signal. The only cable connection to the model had a diameter of approximately 7 mm and was necessary to control the motor of the wind turbine (see Figure 2). Therefore, a negligible contribution of the cable connection to the motion of the model is expected.

## Appendix E: Simulation parameters

*Author contributions.* CWS planned and conducted the experiment, developed the rotor thrust calculation procedure, conducted the validation and prepared the manuscript. SN conducted the simulations, made major contributions to the measurement data processing and reviewed the manuscript. PDK designed, manufactured and characterised the wind generator under the supervision of CWS and SN. MAM supervised all works and reviewed the manuscript.

*Competing interests.* The authors declare no competing interests.

*Acknowledgements.* The authors kindly thank the German Federal Ministry for Economic Affairs and Climate Action (BMWK) for funding the HyStOH project (03SX409A-F) and the ProHyGen project (03EI3084C). Special thanks go to our partners CRUSE Offshore GmbH, aerodyn engineering GmbH, JÖRSS-BLUNCK-ORDEMANN GmbH, DNV-GL and the Institute for Ship Structural Design and Analysis at the Hamburg University of Technology for the excellent cooperation in the HyStOH project. The authors appreciate the work of Uwe Gietz to support the experimental investigations. Furthermore, the authors thank CRUSE Offshore GmbH for supporting the manufacture and design
of the FOWT model.

Publishing fees supported by Funding Programme Open Access Publishing of Hamburg University of Technology (TUHH).

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
