# Peer review of "Low Uncertainty Wave Tank Testing and Validation of numerical methods for Floating Offshore Wind Turbines"

_Wind Energy Science, 2024_

## Author Comment (AC1)

*Dear reviewer,*

*we would like to express our deep thank to you for your careful review of our work. We tried to incorporate most of your comments, which we found generally very helpful and formulated in a positive way. We believe that the manuscript could be improved greatly in this way.*

*We respond to your comments directly in the following:*

This work has a significant scientific contribution, as it deals with a very relevant issue: the testing of floating offshore wind turbines at wave basins. It brings attention to the main uncertainties associated with this type of experiments and proposes some potential improvements on the processes particularly oriented to use of the experimental data for computational tools validation. It also shows how these proposals have been tried out in a particular experiment, showing and analysing the results.

The paper theme is well explained and is supported by referenced previous works. The description of different testing methods is presented highlighting pros and cons. The motivation and objectives of this work are presented in a clear and justified way. The experiment performed is described with detail and the results are shown including a throughout analysis. The writing of the text is clear and well structured.

In general, it is a very well-presented work of great interest due to its subject matter and the results it teaches.

Getting into the different scientific issues addressed in the paper, I have the following comments and questions:

- The wind generator is one of the key aspects for obtaining accurate results on wave basin tests of scaled down floating offshore wind turbines. Obtaining an homogeneous air flow with a wind generator on a non-controlled open space is not straightforward. Description of the wind generator is well performed, and a velocity field measurement is presented at one section.
    - Have these measurements been performed using only a Prandtl probe?
    - How was the turbulence been measured? Any high frequency measuring instrument, as hot wire, has been used.
    - Has cross-flow (y, z directions) been measured?

*A Prandtl probe has been used (see section 4.1). Unfortunately, it was a unidirectional device. A sentence is added to highlight this shortcoming (also in the comparison with other wind generators):*

*"Another shortcoming of the evaluation of the wind speed quality is that flow velocities perpendicular to the main flow direction have not been measured (or evaluated) neither in literature nor in the present case."*

*No additional high-frequency turbulence measurement equipment has been utilised. Although this would have surely been the more accurate way to characterise the wind tunnel, we found it a bit out of scope to run a second test series, because we did not expect to identify a strong influence of high frequent turbulence on the measured rotor thrust in such non-ideal environment.*

- One of the main questions that arises to me is the way how the wind turbine rotor has been scaled. What I understand from the text is that the rotor is not a scaled down version of a real wind turbine rotor, but an existing model rotor designed for wind tunnel tests as described in reference Shultz 2022. Then, there are some questions about how the full-scale rotor is defined:
  - It is stated that the scaling factor for the wind turbine rotor is 150. Does this mean that the full-scale rotor is a direct geometrical scale-up of the model rotor?
  - It is not clear whether tip speed ratio is maintained between full-scale and experiments. It is stated that the tests are performed at a constant rotational speed and different constant wind speeds. May be stating what was this rotational wind speed at the experiments and the equivalent full-scale one would solve the question.
  - It is not clear to me either how the wind speed has been scaled down. From table 1, it seems that it has been scaled down by a factor of 2 (from 10 m/s to 5 m/s). But criterium for this value is not explained.
  - I don't understand on Appendix A, the part of equation A4 that states that $u_{laero}$ is determined by the ratio of aerodynamic and hydrodynamic scaling factors: $u_{laero} = u_{lhidro} (l_{aero} / l_{hydro})$ . May be this can be explained from the answer to previous question.

*Thanks for this hint. We agree that the scaling was not described as clear as we would wished it to be. We have reworked section 3 to clarify the above comments, but also answer them here shortly. In addition, we have included a comparison with two measurement series in the wind tunnel, where the influence of the TSR and the chosen TSR are shown (see section 7.1).*

- *No, the rotor is just a representative for a full scale turbine as it acts similar as today's full scale turbines in terms of thrust and power generation as well as the load distribution along the blades. To achieve this, the rotor design and airfoils need to be adopted to the low Reynolds number regime.*

- *Due to the fact that the turbine is redesigned and does not correspond to one particular full scale turbine, the TSR is not maintained.*

- *Wind speed scaling: We have modified an existing sentence in section 3, which says:"In order to achieve the Froude similarity of the mean thrust (assuming a constant thrust coefficient and tip speed ratio), the wind speed needs to be increased by the factor λ aero /λ hydro in comparison to conventional Froude scaling of the environmental conditions." This means that in the first step, the wind speed is calculated using Froude similarity (sqrt( λ hydro)) and than increased by a factor λ aero /λ hydro to account for the smaller rotor in comparison to conventional Froude scaling. In addition, we extended appendix A with a more detailed explanation. This should also solve the issue stated in the next point. An advice whether this is understandable would be helpful.*

- The results show inaccuracy on the thrust measurements, specially at the case with smallest wave period. It is stated that inaccuracy would come from inertial and weight measurements, as

derived from the waves only tests, but also the thrust measurement introduces some variability. Has this been more deeply analysed?

*The reviewer hits a critical point here. Of course, this could be the case. In order to check this, the sensor uncertainty (see Table 3) and the repetition error of the thrust force measurement in steady conditions was checked (see section 7.1). Both are most likely not the cause of these strong deviations.*

May be a thrust measurement of the rotor with the plataform fixed (to separate this from inertial and gravitational forces measurement) could be done. Also, if thrust measurements of the rotor were performed at wind tunnel at same conditions, that could be an interesting analysis, to see possible effects of the flow quality

*Thank you for this hint. As mentioned above, we added a comparison of such measurements with wind tunnel tests. Although the turbine is not mounted rigidly, but is free floating, only very little motions and a minimal platform pitch angle are monitored by the motion tracing system (see section 7.1).*

- All the results show the surge, heave and pitch behavior of the model. Since it is a single point mooring platform, and therefore has yaw as free degree of freedom, I wonder if you have anlalysed the yaw behavior and its potential influences.

*This is an interesting remark. We checked the yaw positions of the platform an found it to be between +-2° is all cases with only a small drift during the tests but no motion with the wave frequency. In most cases the variation of the position is even lower. In the 'with wind' cases, we observe a slight shift of the mean yaw angle to 1° and only very small deviations from this. The lower deviation from the mean is due to the fact the tower airfoils align support the alignment of the rotor. The small mean yaw offset arises from the rotor torque that causes a slight roll angle. This roll angle, in turn, moves the application point of the rotor thrust force slightly to the side, which then results in a yaw offset. This is a phenomenon we know from full-scale analysis, which can be found in*

*Netzband, S., Schulz, C. W., and Abdel-Maksoud, M.: Self-aligning behaviour of a passively yawing floating offshore wind turbine, Ship Technology Research, 67, 15–25, https://doi.org/10.1080/09377255.2018.1555986, 2020.*

*However, despite this topic is very exciting, we decided to not discuss it in this publication, because it is already quite broad and we were afraid to further confuse the reader with more topics.*

- The platform has an airfoil shaped passive yaw mechanism covering part of the tower. It is not clear whether it is completely laying on the wake of the wind generator.
  - Could you confirm about this point?

*Yes, this is the case. We added a sentence:*
*"The airfoil is fully covered by the wind field\footnote{Is has to be noted that the flow quality near the lower end of the airfoil around the tower is impaired due to boundary effects of the wind generator}"*

- Although the influence should be small, have you been considering it on the full-scale computations?

*We considered the influence via lift and drag coefficients along the tower in the present (model-scale) simulations and also in related full-scale simulations on other publications. However, the influence of the tower is indeed very limited when the platform is aligned with the wind.*

Finally, as particular technical correction I would suggest to show in a figure the axis conventions used to derive the results in terms of motions directions (surge, heave, pitch) and forces directions as well.

*We agree that an overview of the setup including a coordinate system was missing. We added this and in Figure 7 and described the directions in the corresponding text.*

*Best regards and many thanks for your extensive review,*

*Christian W. Schulz on behalf of the authors*

---

## Author Comment (AC2)

*Dear reviewer,*
*we would like to thanks you very much for taking your time to revise this extensive manuscript. We are aware that it contains a lot of information and text, which most likely made reviewing a quite time consuming task. Therefore, we appreciate that you were willing to invest this work and give us advice to improve it.*

*We respond to your comments directly in the following:*

Several improvements to the validation process aiming at the reduction of the uncertainties were proposed and evaluated in this work. The major improvements were the measurement of the rotor thrust force excluding the tower top inertia loads, a significant improvement of 10 the wind field quality in the wave tank and the utilization of hybrid simulations based on the measured platform motions. These steps were applied to wave tank tests of a FOWT utilizing a single point mooring and the subsequent validation of the numerical panel method panMARE. The improvements allowed for a considerable decrease in the random and systematic uncertainty of

the model tests and made a valuable contribution to the distinction between cause and effect regarding the deviations between measurements and simulations.

This paper covers a lot of contents, so the introduction and analysis of each part is not very full.

*We agree on this and tried to 'fill up' the analysis with a number of points so that a more complete and elaborate analysis results. The main changes in this context are:*

*1. More detailed characterisation of the utilised wind generator and analysis of selected wind generators used in comparable experiments (see section 4.1)*

*2. Comparison of wind tunnel measurements of the same rotor with measurements from the present test campaign and the present simulation model (see section 7.1)*
*3. Introduction of two sub-chapters to quantify the uncertainty of the tests in more detail (see sections 6.1 and 6.2)*

The research content of this paper is very meaningful, but it is hoped that the author can find the main expression content of the paper, accurately refine it, and condense it into a journal article with academic value.

*The reviewer hits a critical point here. The fact that a package of actions rather than one or two main actions have been performed to decrease the uncertainty of the whole validation process makes is very difficult to create a red line throughout the paper. Although the different design choices and applied methods cover a wide field, the authors believe that these need to be considered in one work instead of focusing on e.g. the wind generator only. This is due to the fact that the choice of the 'special' scaling methodology  is a prerequisite the construction of the elaborate wind generator and the coverage of the full rotor area with a high quality wind field. However, as this choice comes along with a number of significant shortcomings due to the violation of the scaling rules, the advantages need to be justified with the quality of the presented results, especially regarding the rotor thrust force and its influence on the motions. This, in turn, requires (most of) the other described design decisions and applied methods (e.g. application of the inertia removal procedure, analysis of time series, focus on simple load cases,*

*hybrid simulations), as the increased accuracy in comparison to state-of-the-art validation approaches is needed to demonstrate the effect of the application of the scaling approach and the elaborate wind generator.*

*We fully understand this point and that the reviewer wishes to put more focus on one or two single topics. However, due to the above mentioned reasons, we find it difficult to correspond to this in a radical way without conflicting with the other reviewers, who seemed to be okay with the current general structure or even commented it positively. We therefore added a subdivision of the introduction into introduction, motivation and scope so that the structure of the paper becomes clearer. We are also open for more suggestions on how to practically improve the structure.*

In addition, some suggestions are as follows:

1.The structure of the first and second chapters of the paper is inconsistent with the common articles. The first chapter is basically the author's statement, and the second chapter is the previous research work. In my opinion, the content of the first and second chapters should be properly integrated, and the research review of the second chapter should be used to support the views stated in the first chapter, so as to solve the sense of separation in the first two chapters of the paper.

2.In Chapter 2, so many previous works are introduced, all of which are supported by text. The authors should choose some important content to accompany the figure to illustrate.

*We see the point of the reviewer here (1.), which is most likely in connected to the next comment (2.). The reason why we chose to deviate from the standard way to introduce our work is that we found the literature review to long (and detailed) to be properly integrated into the introduction as both together are about 6 pages long. Unfortunately, we do not know about a well structured work in literature, which discusses the uncertainties of wave tank testing and validation in a general and sufficiently detailed way with respect to the addressed sources of uncertainty in this work. Therefore, we found it necessary to justify our design decisions and evaluation methods on the basis of the issues occurring in original works, which finally leads to a comparatively detailed literature review.*

*As already discussed in the previous answer, we are afraid to impair the quality of the work in the eyes of the other reviewers when applying major changes here. Especially in case of the literature review, we received positive comments on the present version.*

3.The title of Chapter 3 does not summarize the content of Chapter 3 clearly, it is too concise.

*We agree on this and changed it to "Introduction and discussion of the scaling approach"*

4.Chapter 4: It is best to provide a layout diagram here.

*Thanks for hint. We added a layout diagram.*

5.As stated in the abstract: The major improvements: a significant improvement of the wind field quality in the wave tank. This section describes the advantages of this wind generating system, but I think the advantages should be reflected by comparing the quality of the wind generating system with that of other existing basin tests. Therefore, I think a table should be used to compare some important parameters to verify the advantages of this wind generating system.

*Thanks for this hint. We believe that this gives evidence that the choice of this special scaling approach and wind generator design is advantageous with respect to other wind generators. We listed the characteristics of selected wind generators from literature and performed an analysis of the flow quality based on the published figures. This analysis clearly shows the increased flow quality of the present wind generator and test setup (see Table 1 and appendix B) .*

6. Table1: What is the scale ratio of the test?  What scaling criteria do periods, wave heights, and wind speeds follow?

*Thanks for this hint. We forgot to mention that in general, Froude scaling is applied:*

*"While the the platform and the hydrodynamic environment are scaled using conventional Froude scaling with a scaling factor of 45, the geometric scaling factor for the wind turbine rotor diameter was chosen to be  150….. In order to maintain the Froude similarity of the mean thrust assuming a constant thrust coefficient and tip speed ratio, the wind speed needs to be increased by the factor $\lambda_{aero} / \lambda_{hydro}$ in comparison to conventional Froude scaling."*

7. The conclusion is too long, requiring extensive cuts and rigorous generalizations

*The authors agree that the conclusion was not well structured and too long. Major parts have been cut and restructured.*

*Best regards and many thanks for your extensive review,*

*Christian W. Schulz on behalf of the authors*

---

## Author Comment (AC3)

*Dear Mr Goupee,*
*we are very thankful for taking your time to revise this extensive manuscript and for sharing your expertise and experience with us. We found your advice helpful, inspiring and in a positive spirit in all cases and were hopefully able to implement the vast majority of your comments.*

*We respond to your comments directly in the following:*

This reviewer finds this manuscript by C.W. Schulz et al. to be of interest to the FOWT community. The article is also fairly well written and organized.

That said, this reviewer has several comments the authors may wish to consider addressing in a revised version of the manuscript. They are as follows:

1) There is surprisingly little quantitative information on the tested FOWT system (dimensions, mass properties, mooring geometry, etc.). This would appear to make the work difficult to reproduce. There is a mention of some information in Appendix B, but very little can be found there.

*We agree on this. Unfortunately, the platform design is owned by a company. We are therefore not allowed to give more details here.*

2) On a related topic, there is very little specific information on the numerical modeling inputs (e.g., aerodynamic properties, hull drag coefficients, mooring stiffnesses, etc.). Again, without providing this information, the work is difficult to reproduce. Also, adding detail in the modeling inputs may help the reader better understand discrepancies observed between the tests and simulations.

*We added more details to appendix E.*

3) The authors provide a fairly nice literature review, and do a good job of justifying their tank testing choices for the purposes of reducing uncertainty. However, no quantitative uncertainty information is provided for any of the measurements, nor is there any quantitative evidence provided that the approaches employed leader to less uncertainty than competing approaches. Based on the title of this manuscript, this reviewer thinks it is reasonable that a reader may expect this information to be present in the manuscript.

*This is clearly a weakness of the submitted manuscript and we are thankful for this hint. We added a section (6) in incorporated the repeat tests into it. We hope that this provides more evidence for the improvement of the quality of the model tests. In addition, we added a more elaborate evaluation of the wind field quality and a comparison with other wind generators (see section 4.1). In this course, the evaluation of the non-uniformity of the wind generator was performed more precise so that even lower values resulted. Finally, a comparison with a wind tunnel measurement campaign was added under section 7.1 to prove evidence for the high quality of the wind field and sensing system.*

4) Several choices are made to reduce uncertainty in the testing (wind machine design, wind turbine reduced size, focusing on regular waves, etc.). However, these choices deviate somewhat from the physical properties and design load case environments of usual interest in FOWT design. The authors may want to better defend the choice of reducing uncertainty for these 'off-design' scenarios as opposed to attempting more complicated, uncertain, but more realistic tests (closer to Froude-scale turbines, active turbine control, irregular wave environments, etc.). The former may reduce

uncertainty, the latter will provide data that can be used to exercise numerical models in areas more relevant for modern FOWT design. As an example, the environments considered in this work will not induce any second-order wave drift forces which can significantly impact certain FOWT dynamic responses. In addition, these are often the hardest to capture with numerical models, and as such, are of great interest in tank testing campaigns.

*Thanks for this hint. We added a paragraph, which covers most of the points. However, we tried to keep it short in order to avoid lengthening the paper too much.*

*"… Therefore, although the Froude similarity of tower top motion velocity and wind speed is violated, the utility system is well suited for the validation of numerical methods. The special value of the proposed scaling approach is to enable a precisely known wind environment and wind turbine characteristic. This is achieved by a more elaborate wind generator and a much better covering of the (moving) rotor swept area compared to other tests due to the small size of the turbine. The ability to test the smaller rotor in a wind tunnel environment with sufficiently low blockage ratio and under highly controlled conditions (see section \ref{chap:windtunnel}) opens the possibility to validate the aerodynamic simulation model accurately and to identify measurement differences arising from the non-ideal wave tank environment. Both together leads to a well defined (and known) thrust force, which is applied to the tower top, with a comparatively low level of noise. This, in turn, yields a low contribution of the aerodynamic system to the random and systematic uncertainty of the platform motion, which is a major improvement in comparison to the above listed studies. In addition, the low level of noise enables the reliable compensation of gravitational and inertial loads from the tower top forces so that a direct validation of the simulated rotor thrust is possible during a motion cycle. Naturally, these improvements over existing test strategies are achieved in exchange with a less realistic behaviour of the model FOWT as discussed above. "*

5) In Figure 4, the amplitude of the dynamic thrust force is approximately $1/3^{rd}$ of the anticipated full scale amplitude. An explanation and derivation is provided, which is appreciated. However, is the uncertainty in this dynamic thrust force reduced by at least $1/3^{rd}$ as well? If not, perhaps other approaches that can capture not only the mean thrust force, but the full-scale variation in the thrust force would be better to pursue as they better represent the desired physics and the uncertainty as a percentage of variation would be no worse than the proposed approach (consider reviewing some of the other works produced on the recent FOCAL test campaign).

*This is an important note. We believe that we have to distinguish two things here: First, the application of a well defined and realistic rotor thrust to the tower top and second, the ability to measure this load during waves and compare it to simulations.*

*When considering the first point, it might be questionable if the introduced load by the rotor is three times more accurate. However, in the work of Mendoza (2022), it can be seen that the repetition error of the rotor thrust is in a range of 1% or a bit more, which is indeed more then our estimation of 0.6%. In addition, the ability of the numerical models to reproduce the rotor thrust over a certain range of TSRs shown in this work is much lower compared to our newly added figure the present publication. This itself is not an argument, because the numerical models might be erroneous, however, it gives a*

*hint that some physical effects like uncertainty in the wind field, the measurement system or the blade geometry also be present.*

*From our point of view, the second point is more important: The aim of the testing is to identify modelling inaccuracies in particular part of the model. Therefore, the ability to measure the aerodynamic rotor thrust force while the platform undergoes motions is a key capability of this way of testing as this allows for a direct comparison with the aerodynamic simulations. We reviewed some of the the FOCAL-related works again and did not find a comparison between simulated in measured aerodynamic rotor thrust when the turbine undergoes wave motions. (And no quality check of a potential inertia and gravitational load compensation.) We are also not aware of any other successfully validated attempt to do so. Please let us know is you are aware of something like this. Considering our state of knowledge as given, a comparison of the uncertainty of rotor thrust measurements would not even possible because no other approach proofed to be working. Of course, this stands under the retention of the existence of other works. Please let us know if you know about such works. In this case, we would naturally compare the uncertainties to each other.*

6) Several qualitative descriptions of the size of the wind machine relative to the wind turbine are provided. Consider precisely quantifying the room the rotor has to move in heave/sway while still remaining in the low spatial variation, low turbulence intensity portion of the wind machine jet.

*We added a sentence:*

*"Deviations in oft the rotor position in heave and sway direction of approximately 0.2\,D are tolerable in this context. This limit is by far not exceeded in the present tests."*

7) In Figure 6, it would be nice if the color bar variation was focused more on the rotor area; by including the pieces outside of the jet, it is hard to visually pick up on the turbulence intensity and spatial variation trends in the rotor plane.

*We agree on this and changed the figure.*

8) The last paragraph on page 14 discusses a modeling approach where an angle of attack offset is used as a viscous correction. Is this a standard approach? If so, can a reference be provided? This reviewer has not seen this method used before in a model correlation study.

*This method is more or less unique because it is only relevant for panel methods and for the special situation, where we have a very narrow band of angles of attack at the rotor radial stations. The method is described a bit more detailed in*

*Schulz, C. W., Netzband, S., Özinan, U., Cheng, P. W., and Abdel-Maksoud, M.: Wind turbine rotors in surge motion: new insights into unsteady aerodynamics of floating offshore wind turbines (FOWTs) from experiments and simulations, Wind Energy Science, 9, 665–695, https://doi.org/10.5194/wes-9-665-2024, 2024*

*However, this is not relevant any more, because we decided to replace the panel method with our inhouse lifting-line method. This is due to the fact that we now have a LL model of the exact same rotor had to rerun all the cases when we incorporated the 'real' waves. The results are only slightly deviated by this change and the analysis is adjusted slightly.*

9) No information is given on conditions to reach the 'steady-state' responses shown in the plots, nor how many cycles are included in the plots.

*Thanks for this hint. We forgot to include this. The following sentence was added:*

*"For this and all following analyses, phase averaged data was computed on the basis of measurements, which started after aperiodic effects decayed, which took at least ten motion cycles after the approaching of the first wave. The data sets had a length of 6 - 12 motion cycles, depending on the quality of the generated waves and the occurrences of obstacles or noise in certain measurement channels"*

10) This reviewer has seen other works that perform 'hybrid' simulations of mooring systems with better comparisons between experiment and simulation. The quality of the comparisons depends significantly on the numerical approach being employed in the mooring analysis (quasi-static, lumped mass, FEM, etc.) and mooring line hydrodynamic properties (e.g., transverse drag coefficient). The author M. Hall that has been referenced in this work has a good article that may be worth reviewing.

*We are thankful for this hint. We did not know about this particular part of Hall's work. This demonstrates nicely the idea that 'hybrid simulations' are of major value when identifying the source of particular mismatches between simulations and experiments. We added a reference to hist work. Indeed, our implementation of the mooring model is tightly oriented on the model Hall proposes in*

*"Validation of a lumped-mass mooring line model with DeepCwind semisubmersible model test data".*

*However, we are aware of the sensitivity of the results on the choice of the mooring system parameters and could not find a suitable set of parameters, which leads to much better results. Therefore, we believe that our mooring system itself contains too complex effects (in comparison to real mooring systems) to be captured accurately. This view can be supported by considering the amount of high frequent oscillations and obstacles in the measured mooring loads. In addition, it has to be noted that Hall compares overall the mooring line tension, which is most likely dominated by the load in heave direction due to the angle of the mooring lines. In this direction, our simulations also show fair agreement, while the large differences occur in surge direction, which is not directly evaluated in Hall's work.*

11) For the full simulations, why aren't the actual recorded waves used as inputs? Without doing so, it is hard to understand which discrepancies are due to modeling deficiencies/test uncertainties, and which are due to incorrect model inputs. This reviewer thinks it is common practice to use the as measured wave in model correlation studies, and am surprised that this is not done here.

*We agree on this. Some software problems prevented us from doing so. However, these problems have been solved in the meantime and we repeated all simulations with the 'real' waves. All figures have been exchanged and the analysis was slightly reworked. However, the agreement between measurements and simulations did not improve significantly.*

12) Is there a reason the results are not provided at full scale? Results are usually more intuitive when presented at full scale in this reviewer's opinion.

*Although it is common to present upscaled data, we decided to stick to real values. We did this for a number of reasons:*

- *The simulations have been performed in model-scale as the viscous correction shall work in the correct Reynolds number regime. We do not want to suggest something else tot he reader.*

- *Model and sensing system uncertainties as well as estimations on repetition errors are given in absolute numbers. For us as researchers that deal with this kind of measurement equipment and data, absolute values are way more intuitive. In order to compare these uncertainties with the data, the scale should be the same.*

- *We decided to not normalise the rotor thrust force in terms of a Ct value as it should stay directly comparable to the mooring surge force.*

- *Due to the variety of turbine ratings used for FOWT analyses in literature (~2 – 22 MW), we do not find absolute values for heave, surge, rotor thrust and mooring loads at scale much more intuitive compared to the model scale data as those strongly depend on the full-scale turbine rating.*

13) There are some other minor issues that should be addressed: There are some widows/orphans, the paragraph indenting is inconsistent, and the figures are often not located very closely to their first mention in the text.

*Thanks for the hint. We tried to fix the inconsistencies in indentation. As the typesetting will change strongly when the paper is converted to the journal layout, we will fix the remaining issues at this stage.*

*Best regards and many thanks for your extensive review,*

*Christian W. Schulz on behalf of the authors*